# Using Monte Carlo conformal prediction to evaluate the uncertainty of deep learning soil spectral models

Yin-Chung Huang, José Padarian, Budiman Minasny, and Alex B. McBratney

School of Life and Environmental Science & Sydney Institute of Agriculture, The University of Sydney, NSW, Australia

5    *Corresponding to:* Yin-Chung Huang (lloyd.yc.huang@sydney.edu.au)

**Abstract.** Uncertainty quantification is a crucial step for the practical application of soil spectral models, particularly in supporting real-world decision making and risk assessment. While machine learning has made remarkable strides in predicting various physiochemical properties of soils using spectroscopy, its practical utility in decision-making remains limited without quantified uncertainty. Despite its importance, uncertainty quantification is rarely incorporated into soil spectral models, with existing methods facing significant limitations. Existing methods are either computationally demanding, fail to achieve the desired coverage of observed data, or struggle to handle out-of-domain uncertainty. This study introduces an innovative application of Monte Carlo conformal prediction (MC-CP) to quantify uncertainty in deep learning models for predicting clay content from mid-infrared spectroscopy. We compared MC-CP with two established methods: (1) Monte Carlo dropout and (2) conformal prediction. Monte Carlo dropout generates prediction intervals for each sample and can address larger uncertainties associated with out-of-domain data. Conformal prediction, on the other hand, guarantees ideal coverage of true values but generates unnecessarily wide prediction intervals, making it overly conservative for many practical applications. Using 39,177 samples from the mid-infrared spectral library of the Kellogg Soil Survey Laboratory to build convolutional neural networks, we found that Monte Carlo dropout itself falls short in achieving the desired coverage – its 90 % prediction intervals only covered the observed values in 74 % of cases, well below the expected 90 % coverage. In contrast, MC-CP successfully combines the strengths of both methods. It achieved a prediction interval coverage probability of 91 %, closely matching the expected 90 % coverage and far surpassing the performance of the Monte Carlo dropout. Additionally, the mean prediction interval width for MC-CP was 9.05 %, narrower than conformal prediction's 11.11 %. The success of MC-CP enhances the real-world applicability of soil spectral models, paving the way for their integration into large-scale machine-learning models, such as soil inference systems, and further transforming decision-making and risk assessment in soil science.

## 1 Introduction

In the recent developments of soil science, machine learning has been widely used in applications such as soil spectroscopy, proximal sensing, carbon stock modelling, and digital soil mapping (Ng et al., 2019; Wadoux et al., 2020). These studies are characterised by the use of large soil datasets and require an efficient way of extracting information to predict target attributes. Hence, machine learning is favoured because these algorithms can generate prediction models with high accuracy for various purposes (Padarian et al., 2020; Minasny et al., 2024). For example, in soil spectroscopy, visible and near-infrared (Vis-NIR) and mid-infrared (MIR) spectroscopy have been used with machine learning to predict soil properties such as soil organic carbon (SOC), texture, and cation exchange capacity (CEC). Padarian et al. (2019) applied convolutional neural networks (CNN) to predict soil properties with Vis-NIR spectra, using the European Land Use/Cover Area Frame Statistical Survey (LUCAS) dataset, which contained about 20,000 samples. In the SOC prediction of their study, the multi-task CNN outperformed conventional algorithms, such as partial least squares regression and Cubist, by reducing the root mean square error (RMSE) by more than 60 %. Additionally, Ng et al. (2019) used the Kellogg Soil Survey Laboratory (KSSL) database with around 15,000 samples from the United States (US) with both Vis-NIR and MIR spectra to build a multi-task CNN. Their model achieved a coefficient of determination ($R^2$) over 0.90 for total carbon, SOC, CEC, clay, sand, and pH.

Despite the significant success of machine learning in predicting soil properties, uncertainty quantification of the prediction remained an underexplored area in soil spectroscopy (Omondiagbe et al., 2024). The growing demand for practical applications of soil spectral models requires users to know the uncertainty accompanying the model prediction to assess the quality of the predictions (Bellon-Maurel et al., 2010). Additionally, deep learning (DL), as a branch of machine learning, is increasingly being applied to soil science to explore its ability to extract information from large datasets. In the data-intensive context of deep learning, uncertainty analysis is critical in evaluating models for decision-making and risk management, and predictions without uncertainty are neither practicable nor applicable (Begoli et al., 2019). Hence, it is crucial to establish an effective way to evaluate the uncertainty of machine learning models.

An ideal uncertainty quantification method is expected to satisfy the following criteria:

    (1)   the method is computationally efficient,

    (2)   the prediction interval coverage probability (PICP) must meet the expected coverage, i.e. a $p$ % coverage is expected for a $p$ % prediction interval, with the narrowest mean prediction interval width (MPIW), and

    (3)   the prediction intervals should be able to address the larger uncertainty for samples significantly different from the training set (i.e. out-of-domain samples).

Several methods have been used to generate intervals for each prediction to characterise uncertainty. One commonly used approach is bootstrapping, in which several models are trained with subsets generated by drawing samples with replacements from the same dataset (Efron and Tibshirani, 1994). The mean of all models is considered the final prediction, and an interval can be derived from the quantiles of multiple predictions. However, one drawback of bootstrapping is the time-consuming nature of training numerous bootstrapping models. In addition, bootstrapping primarily addresses the model uncertainty and it derives confidence intervals rather than prediction intervals (Heuvelink, 2014; Wadoux, 2019). A comprehensive uncertainty quantification using methods such as Markov Chain Monte Carlo can better evaluate the parameter uncertainty involved in the model (Minasny et al., 2011).

The diverse nature of models enabled the development of different methods. For example, quantile regression (QR) uses a set of regression models to estimate the quantile of target variables, and the prediction interval can later be defined by the upper and lower quantiles (Kasraei et al., 2021). Additionally, quantile regression forests (QRF) and quantile regression neural networks (QRNN) are also extensions of quantile regression that apply similar principles to generate prediction intervals (Schmidinger and Heuvelink, 2023). Heuvelink et al. (2021) utilised QRF to predict the SOC for soils in Argentina with quantified uncertainty, and the 0.05 and 0.95 quantiles were used to generate the 90 % prediction interval. However, QR is not yet available for every DL model. On the other hand, Omondiagbe et al. (2024) compared bootstrapped PLS, generalised additive models (GAM), and Bayesian CNNs for their ability to quantify uncertainty. They found that GAM and Bayesian CNN outperformed bootstrapped PLS by having PICP close to the ideal 90% value. Moreover, the MPIW of Bayesian CNN is mostly lower than that of GAM models, suggesting a more accurate estimation of uncertainty (Omondiagbe et al., 2024). However, Bayesian neural networks are more intensive in computation compared to standard CNNs (Bethell et al., 2024; Omondiagbe et al., 2024).

An alternative method to evaluate model uncertainty in DL is the Monte Carlo dropout (MC dropout) by Gal and Ghahramani (2016), in which a CNN model is trained with multiple dropout layers that randomly deactivate neurons during prediction, resulting in different predictions across iterations. Multiple predictions from a single MC dropout CNN model form a distribution, and prediction intervals could be obtained by assessing the quantiles of the predictions. This approach reduced the training time compared to bootstrapping.

The performance of bootstrapping and MC dropout was compared by Padarian et al. (2022), in which CNN models were trained to predict SOC with Vis-NIR spectra using the LUCAS dataset through (1) a hundred times bootstrapping and (2) MC dropout. Additionally, CNN models were trained on mineral soils with a threshold of <20 % SOC and then separately tested on in-domain data (mineral soils, SOC<20 %) and out-of-domain data (organic soils, SOC>20 %). This was to test the model's

response to samples significantly different from the training set. A good uncertainty quantification should indicate the larger uncertainty when predicting out-of-domain data. When facing in-domain data, both bootstrapping and MC dropout generated reasonable prediction intervals. However, when facing out-of-domain data, the prediction interval of MC dropout increased significantly compared to bootstrapping, indicating that the uncertainty increased when the testing samples were markedly different from the training data. In other words, the model was aware of its uncertainty for out-of-domain data and can reflect this situation by generating a wider prediction interval. Such analysis is particularly useful when assessing risk management, as predictions with higher uncertainty must be treated cautiously. However, both bootstrapping and MC dropout underestimated the uncertainty and were overconfident in their study. The 90 % PICP of bootstrapping and MC dropout in their study were both under 80 % while the expected coverage was 90 %. This was not practical in real-world situations and left room for improvement.

A relatively easier method to generate prediction intervals with expected coverage is conformal prediction (CP), which uses an independent calibration set to estimate the prediction interval and can be performed on any model (Shafer and Vovk, 2008). CP can therefore be integrated with methods such as QR and MC dropout. Kakhani et al. (2024) utilised CP to generate prediction intervals for SOC mapping in Europe with the LUCAS dataset, and found CP outperformed other methods by generating the most accurate PICP and a reasonably-sized prediction interval. Singh et al. (2024) applied CP with ML in earth observation data, and CP successfully generated prediction intervals of canopy height. Despite these advantages, a key limitation of CP is its inability to generate sample-specific prediction intervals. Instead, it produces a uniform interval for all samples. In other words, CP does not account for increased uncertainty in out-of-domain samples. As a result, CP is known as a conservative method that provides overly broad prediction intervals. This empirical method is similar to the UNEEC (uncertainty estimation based on local errors and clustering) method of Solomatine and Shrestha (2009). UNEEC derived the upper and lower prediction intervals based on the distribution of model error grouped by predictors. Malone et al. (2011) modified the UNEEC method to deal with our-of-domain predictions using fuzzy k-means with extragrades. However, the same as CP, the method is highly dependent on the training data. Consequently, no uncertainty quantification method applied in soil spectroscopy has yet combined computational efficiency, expected coverage with a narrow MPIW, and the ability to address out-of-domain uncertainty.

In this study, we applied the Monte Carlo-Conformal Prediction (MC-CP), a method introduced by Bethell et al. (2024) to improve the PICP of MC dropout while maintaining its advantages in characterising out-of-domain uncertainty. Also known as Conformalised Monte Carlo Prediction, MC-CP not only retains the structure of the MC dropout to generate different prediction intervals for each sample but also extends the prediction interval with CP to achieve the expected coverage. In other words, MC-CP can ensure expected coverage while accounting for the uncertainty associated with each sample. Bethell et al. (2024) demonstrated the effectiveness of MC-CP in both regression and classification tasks using benchmark datasets and showed that MC-CP was significantly improved from the original MC dropout. Hence, MC-CP is a promising method for soil science and can address the uncertainty involved in prediction using DL models.

This study aimed to explore the use of MC-CP as a potential method to quantify the uncertainty of DL models in soil spectroscopy. Specifically, the goal was to validate whether MC-CP preserves the advantages of both MC dropout and CP. Therefore, the objectives of this study are to (1) test if MC-CP can generate prediction intervals that reach expected PICP, and (2) evaluate if MC-CP can address the uncertainty of out-of-domain samples.

## 2 Materials and methods

### 2.1 Dataset

The soil samples from the Kellogg Soil Survey Laboratory (KSSL) dataset were used in this study. It contained the MIR spectra and physiochemical properties of over 17,000 soil profiles and 70,000 soil samples across the US (Soil Survey Staff,

2014). Soil clay content was selected as the target variable to predict with MIR in this study, as the prediction of clay has been a well-established method for MIR spectroscopy (Seybold et al., 2019; Ng et al., 2022). The database contains 45,339 samples
which have measured MIR spectra and particle size analysis. Since the spectra of mineral and organic soils behave differently, samples with SOC > 10 % were excluded, resulting in the removal 1,808 samples. Additionally, extreme values for clay content were also filtered by excluding data below the 5[th] percentile and above the 95[th] percentile, further removing 4,354 samples. This resulted in a total number of 39,177 samples.

Here we created a model based on the in-domain data, and a threshold of 40 % clay content was chosen to separate the in-
130 domain and out-of-domain samples. A clay content of 40 % is the minimum threshold for a soil to be classified as "clay" according to the US Department of Agriculture soil texture classification (Soil Science Division Staff, 2017). Using this criterion, approximately 10% of the samples were categorised as out-of-domain (clay>40 %, N=3,686), while the remaining in-domain samples had clay content below 40% (N=35,491). The in-domain samples would be used for model training, validation, and testing. The in-domain data were further randomly separated into 85 % training, 5 % validation, 5 % calibration
for conformal prediction, and 5 % testing. Only the training and validation data were used in building the model. The out-of-domain samples were not involved in any of the training processes and were only used to test the performance of models when facing out-of-domain situations.

The MIR spectra in the range of 4000-600 cm[-1] were used to predict the clay content. The full procedure of MIR spectral analysis can be found in the manual by Soil Survey Staff (2014). No other preprocessing was applied to the raw spectra as it
has been proven that CNNs are able to deal with spectra without preprocessing (Ng et al., 2019; Padarian et al., 2019). To train the CNN model, the clay contents were scaled to a range of 0-1 using the maximum and minimum of the training set (see Section 2.3).

## 2.2 Uncertainty quantification methods

### 2.2.1 Monte Carlo Dropout (MC dropout)

MC dropout was introduced by Gal and Ghahramani (2016) based on dropout layers, which are commonly used in DL models to prevent overfitting (Srivastava et al., 2014). In each dropout layer, a certain portion of the neurons is randomly deactivated (weights set to zero) during both training and testing. By randomly dropping neurons and their connections, the dropout layer helps prevent the model from overfitting the training dataset. As a result of the dropout layers, each prediction result is different, and multiple predictions generate a distribution. Gal and Ghahramani (2016) demonstrated that MC dropout can be used to
approximate Bayesian inference in deep Gaussian processes, and the standard deviation of the prediction can thus be used for assessing the uncertainty (Bethell et al., 2024). For a detailed rationale, readers are referred to the paper by Gal and Ghahramani (2016).

In practice, a CNN model with dropout layers was trained and performed 100 forward passes with dropout layers activated to generate a predictive distribution (Bethell et al., 2024). In Eq. 1, $X_i$ represents an individual input sample. The 90 % prediction
interval of MC dropout ($C_{MC, 90}$) of each sample $i$ would be defined by the 5[th] quantile ($\hat{q}_5(X_i)$) and the 95[th] quantile ($\hat{q}_{95}(X_i)$) of the predictions (Eq. 1):

$$C_{MC,90}(X_i) = [\hat{q}_5(X_i), \ \hat{q}_{95}(X_i)]$$ (Eq. 1)

### 2.2.2 Conformal Prediction (CP)

CP is a model-agnostic method, which means it can be used to evaluate the uncertainty of any model (Shafer and Vovk, 2008). Consider $(X_i, Y_i)$, i = 1, 2, … n to be pairs of features (inputs) and responses (outputs), and $\alpha$ is the desired error level. A regression model $f$ is constructed using the training dataset, and $f(X_i)$ is the prediction of the observed value $Y_i$. The goal is

to generate prediction intervals $C(X_i)$ such that the probability of the observed value $Y_i$ being contained within $C(X_i)$ is approximately $1 - \alpha$ (Angelopoulos and Bates, 2022). The procedure can be separated into three steps:

(1) **Start with nonconformity scores.** The nonconformity measure is the foundation of CP, which quantifies the difference between the predicted values and the observed values (Shafer and Vovk, 2008). In a regression scenario, the nonconformity measure is typically defined as the absolute value of residuals $r_i = |f(X_i) - Y_i|$. Here, $r_i$ represents the nonconformity scores of the $i$-th data point. The first step is to calculate these nonconformity scores using a calibration dataset and rank the nonconformity scores from low to high. Table 1 shows an example dataset of 100 samples with the $r_i$ in the order from minimum to maximum.

**Table 1: Example dataset of conformal prediction containing 100 samples. The nonconformity scores are ranked from minimum to maximum.**

| $N$ | $f(X_i)$ | $Y_i$ | $r = \|f(X_i) - Y_i\|$ (Nonconformity scores) | $C(X_i) = [f(X_i) - \hat{q}, f(X_i) + \hat{q}]$ |
|---|---|---|---|---|
| 1 | 96 | 95.9 | 0.1 | [93.8, 98.2] |
| 2 | 3 | 3.2 | 0.2 | [0.8, 5.2] |
| 3 | 96 | 95.7 | 0.3 | [93.8, 98.2] |
| 4 | 18 | 18.4 | 0.4 | [15.8, 20.2] |
| 5 | 71 | 70.5 | 0.5 | [68.8, 73.2] |
| 6 | 99 | 99.6 | 0.6 | [96.8, 101.2] |
| 7 | 38 | 37.3 | 0.7 | [35.8, 40.2] |
| 8 | 11 | 11.8 | 0.8 | [8.8, 13.2] |
| 9 | 74 | 73.1 | 0.9 | [71.8, 76.2] |
| 10 | 54 | 55 | 1.0 | [51.8, 56.2] |
| | | | ... | |
| 91 | 24 | 21.9 | 2.1 | [21.8, 26.2] |
| 92 | 56 | 58.2 | **2.2** | [53.8, 58.2] |
| 93 | 48 | 45.5 | 2.5 | [45.8, 50.2] |
| 94 | 19 | 21.8 | 2.8 | [16.8, 21.2] |
| 95 | 90 | 86.9 | 3.1 | [87.8, 92.2] |
| 96 | 27 | 30.2 | 3.2 | [24.8, 29.2] |
| 97 | 70 | 66.6 | 3.4 | [67.8, 72.2] |
| 98 | 66 | 69.9 | 3.9 | [63.8, 68.2] |
| 99 | 21 | 16.8 | 4.2 | [18.8, 23.2] |
| 100 | 80 | 84.5 | 4.5 | [77.8, 82.2] |

(2) **Get adjusted quantile.** Using the ranked nonconformity scores, CP computes the adjusted quantile to determine the prediction interval. Specifically, it selects the $\frac{\lceil (1-\alpha)(n+1) \rceil}{n}$-th quantile of the nonconformity scores $r_i$ to be $\hat{q}$. The $\lceil \; \rceil$ symbol indicates the ceiling function, and this equation is to correct the quantile for the size of the calibration dataset (Angelopoulos and Bates, 2022). In the example, if we set $\alpha = 0.1$ with a total of 100 samples, the $\hat{q}$ will be the 92$^{nd}$ quantile of $r_i$, which is 2.2 (marked as bold in Table 1).

(3) **Generate prediction intervals.** The prediction intervals are constructed as Eq. 2:

$$C_{CP}(X_i) = [f(X_i) - \hat{q}, \; f(X_i) + \hat{q}] \tag{Eq. 2}$$

The width of each prediction interval is fixed to two times the value of $\hat{q}$, centred around the model prediction $f(X_i)$. In the example from Table 1, the prediction interval covers the observed values from sample 1 through 92, indicating 92 % of the samples are covered within the prediction interval. This $\hat{q}$ will be applied to the testing set to generate prediction intervals for unknown data. The key advantage of CP is that it can be applied to any model, regardless of correctness, assumptions, or structure of the model while providing guaranteed coverage for the specified confidence level (Angelopoulos and Bates, 2022). However, the fixed interval width for all data points and guaranteed coverage also makes CP an over-conservative method that generates unnecessarily wide intervals (Bethell et al., 2024).

**2.2.3 Monte Carlo-Conformal Prediction (MC-CP)**

MC-CP is a novel uncertainty quantification method developed by Bethell et al. (2024). As its name suggests, MC-CP combines MC and CP to estimate the uncertainty. Instead of using CP to generate a prediction interval, MC-CP extends the prediction interval from an MC method. The original paper by Bethell et al. (2024) used deep quantile regression to generate prediction intervals, while this study introduces the CNN with dropout layers. The CNN model with dropout layers was trained in the same way as the MC dropout method to predict the calibration set 100 times. For each sample $i$ in the calibration set the 5$^{th}$ quantile ($\hat{q}_5(X_i)$) and the 95$^{th}$ quantile ($\hat{q}_{95}(X_i)$) of the 100 predictions are calculated, and the nonconformity score $E_i$ is defined as Eq. 3:

$$E_i \coloneqq Max\{\hat{q}_5(X_i) - Y_i, \; Y_i - \hat{q}_{95}(X_i)\} \tag{Eq. 3}$$

According to Eq. 3, the nonconformity scores are calculated as the largest distance between the observed value and the boundary of the MC dropout interval. The $\frac{\lceil (1-\alpha)(n+1) \rceil}{n}$-th quantile of the nonconformity scores $E_i$ will then be selected as $\hat{Q}$. The adjusted prediction interval of MC-CP will be calculated as Eq. 4:

$$C_{MC-CP,90}(X_i) = \left[\hat{q}_5(X_i) - \hat{Q}, \; \hat{q}_{95}(X_i) + \hat{Q}\right] \tag{Eq. 4}$$

In MC-CP, the prediction interval of the MC dropout method will be extended by two times $\hat{Q}$. For unknown testing data, the prediction interval will first be calculated in the same way as the MC method and then extended by two times $\hat{Q}$, which is calculated from the calibration set. This will result in sample-dependent prediction intervals, guaranteed coverage, and less conservative intervals than CP.

**2.3 Model architecture and training data**

A 1D CNN was constructed with five trainable layers, namely four convolutional layers and one fully connected (dense) layer. A detailed description of the layers is presented in Table 2. A fixed filter size of five was used for all convolutional layers, and

the filter size for the max-pooling layer was fixed at two. The number of filters started at 32 and increased to 256. Every convolutional layer was followed by a max-pooling layer and an MC dropout layer, resulting in a total of four dropout layers with a fixed 20 % dropout rate. Dropout rates, including 10 %, 20 %, and 30 %, were tested and optimised. The network was trained with a batch size of 300, a maximum number of epochs of 500, and early stopping on the validation set, with a patience of 60. The initial learning rate was set to 0.001, and the learning rate reduction factor was set to 0.1, with a patience of 50. These hyperparameters were also tested and optimised for this dataset. All the analyses were performed in Python v3.12.3 using Tensorflow v2.16.1 (Abadi and Zheng, 2015; Python Software Foundation, 2024).

**Table 2: Architecture of the convolutional neural network. ReLU stands for rectified linear unit.**

| Layer type | Filter size | Filters | Activation |
|---|---|---|---|
| Convolutional | 5 | 32 | ReLU |
| Max-Pooling | 2 | | |
| MC Dropout (0.2) | | | |
| | | | |
| Convolutional | 5 | 64 | ReLU |
| Max-pooling | 2 | | |
| MC Dropout (0.2) | | | |
| | | | |
| Convolutional | 5 | 128 | ReLU |
| Max-pooling | 2 | | |
| MC Dropout (0.2) | | | |
| | | | |
| Convolutional | 5 | 256 | ReLU |
| Max-pooling | 2 | | |
| MC Dropout (0.2) | | | |
| | | | |
| Flatten | | | |
| Fully-connected | | | Linear |

### 2.4 Model evaluation

The model performance was evaluated using coefficients of determination ($R^2$, Eq. 5) and root mean squared error (RMSE, Eq. 6) (Ng et al., 2022):

$$R^2 = 1 - \frac{\sum_{i=1}^{n}(y_i - \hat{y}_i)^2}{\sum_{i=1}^{n}(y_i - \bar{y}_i)^2} \tag{Eq. 5}$$

$$RMSE = \sqrt{\frac{\sum_{i=1}^{n}(y_i - \hat{y}_i)^2}{n}} \tag{Eq. 6}$$

The results of uncertainty quantification were evaluated using the prediction interval coverage probability (PICP, Eq. 7) and the mean prediction interval width (MPIW, Eq. 8) following (Shrestha and Solomatine, 2006):

$$PICP = \frac{1}{n} count \, j \tag{Eq. 7}$$

$$j: PL_i^L \leq y_i \leq PL_i^U$$

$$MPIW = \frac{1}{n}\sum_{i=1}^{n}[PL_i^U - PL_i^L] \tag{Eq. 8}$$

where $n$ is the total number of observations, and $j$ is the number of samples in which the observed value $y_i$ is covered in the prediction interval. $PL_i^L$ and $PL_i^U$ are the lower and upper bounds of the prediction interval of the $i$-th sample. PICP calculates the proportion that the true value is covered by the interval, while MPIW calculates the average length of prediction intervals.

## 3 Results and discussion

### 3.1 Model performance

The DL model demonstrated a good performance in predicting the clay content of the in-domain test set (clay < 40 %), with $R^2$ of 0.90 and RMSE of 3.39 % (Table 3). The results were comparable to those of the multi-task CNN models by Ng et al. (2019), which used part of the current dataset. For out-of-domain samples, a negative R-squared value indicates that the model performs worse than simply using the mean prediction (Fig. 1; Table 3). Such a result for out-of-domain samples was expected, as the model lacked knowledge of soils with clay content exceeding 40 %, resulting in the most out-of-domain predictions falling below 40 % clay.

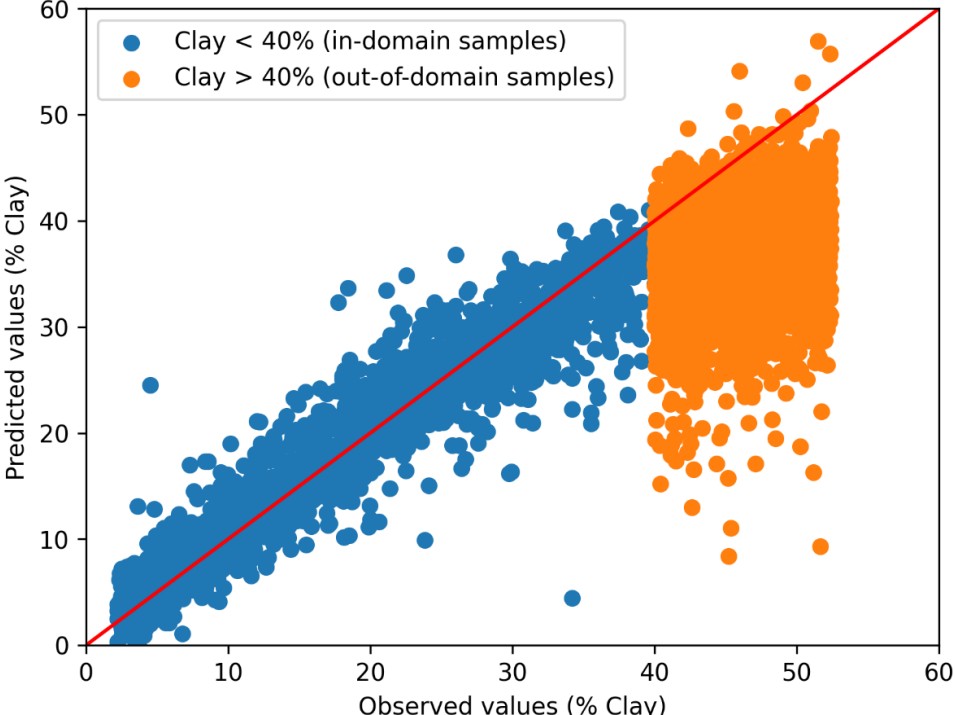

**Figure 1: Relationship between the observed and predicted clay content (%) of the convolutional neural network model for in-domain testing set and out-of-domain samples.**

**Table 3: Results of the convolutional neural network modelling. $R^2$ stands for coefficient of determination, and RMSE stands for root mean square error.**

|  | In-domain test set (n=1775) | Out-of-domain (n=3686) |
| --- | --- | --- |
| $R^2$ | 0.90 | -6.64 |
| RMSE (%) | 3.39 | 9.65 |

## 3.2 Uncertainty quantification

Uncertainty quantification serves as a means of evaluating for the prediction intervals. When a model was predicting with higher uncertainty (in the case of out-of-domain samples), the models are expected to generate wider MPIW to indicate its lack of knowledge. Padarian et al. (2022) demonstrated that MC dropout possessed the ability to "know what they know" and produced prediction intervals for out-of-domain samples five times larger than in-domain samples.

Prediction intervals were generated by making 100 predictions of each sample (Fig. 2), and PICP refers to the probability that this interval covers the observed value. The expected coverage of a $p$ % prediction interval is $p$ %, which is indicated by the dotted line in Fig. 3 (Shrestha and Solomatine, 2006). In the current study, the MC dropout continuously underestimated the uncertainty through all prediction intervals (Fig. 3). This trend was similar to the finding of Padarian et al. (2022). In contrast, the PICP of CP and MC-CP for in-domain samples were both close to the expected coverage (Fig. 3). This is attributed to the "guaranteed coverage" features of CP, and MC-CP provides an augmented effect. However, the PICP for out-of-domain samples was low. This was because the CNN model lacked information about the out-of-domain samples.

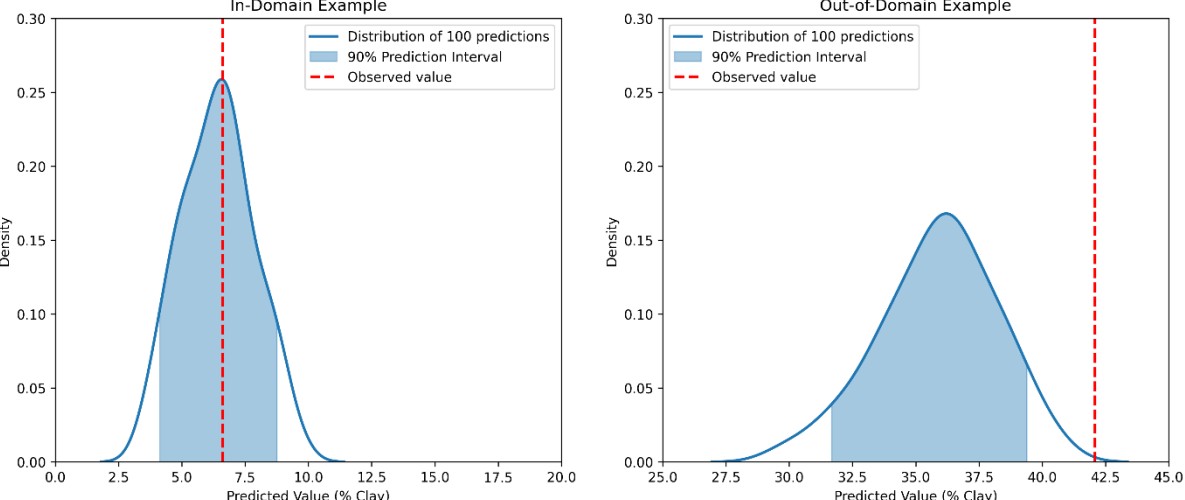

**Figure 2: Examples of the distribution of 100 predictions for an in-domain and an out-of-domain sample using MC dropout. Shaded areas are the 90 % prediction interval. The 90 % prediction interval of the in-domain example covered the observed value while the 90 % prediction interval of the out-of-domain example did not cover the observed value.**

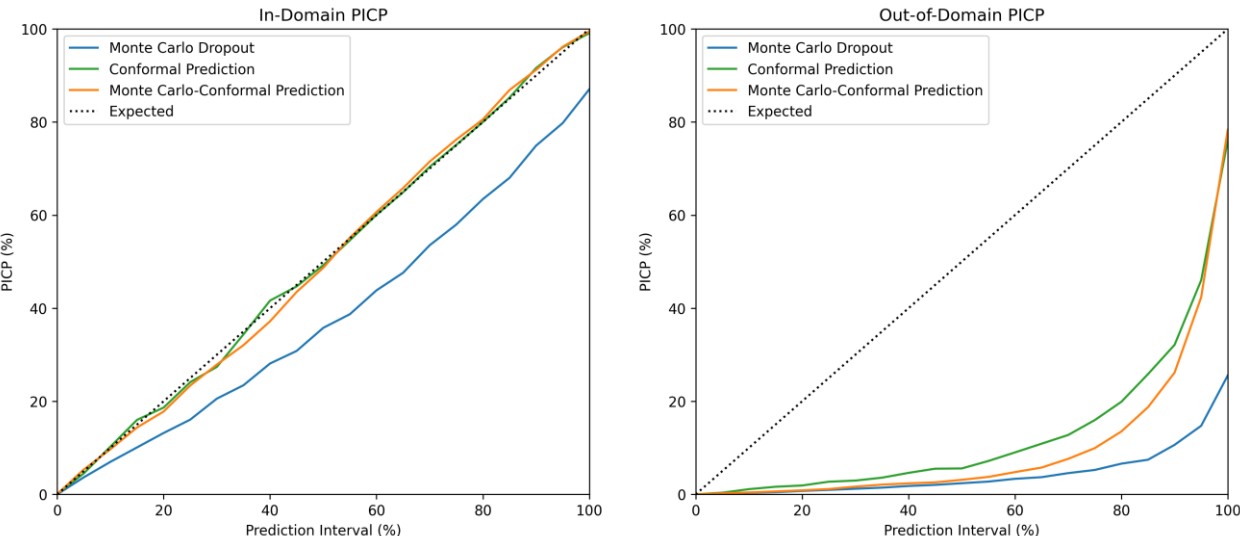

**Figure 3: Prediction interval coverage probability (PICP) of in-domain and out-of-domain samples at different prediction intervals for Monte Carlo dropout, conformal prediction, and Monte Carlo-conformal prediction.**

For 90 % prediction intervals, the MC dropout method achieved 74 % coverage for in-domain samples (Table 4), indicating an overconfident interval. The MPIW of MC dropout for in-domain testing samples was 5.56 %, the narrowest of all three methods (Table 4). This is further supported by the distribution of MPIW in Fig. 4. When encountering out-of-domain samples, the MPIW of MC dropout was 6.93 %, 25 % higher than the MPIW of in-domain samples. This demonstrated the ability of MC dropout to generate wider intervals when encountering samples they are not familiar with (Padarian et al., 2022). However, the extended MPIW was insufficient to fully address the differences between out-of-domain samples and the in-domain training samples, and the 90 % PICP for MC dropout was only 11 %

**Table 4: Results of uncertainty quantification by Monte Carlo dropout, conformal prediction, and Monte Carlo-conformal prediction. PICP stands for prediction interval coverage probability, and MPIW stands for mean prediction interval width.**

| Method | 90 % PICP in-domain | MPIW (%) in-domain | 90 % PICP out-of-domain | MPIW (%) out-of-domain |
|---|---|---|---|---|
| Monte Carlo dropout | 74 % | 5.56 | 11 % | 6.90 |
| Conformal prediction | 91 % | 11.11 | 32 % | 11.11 |
| Monte Carlo-conformal prediction | 91 % | 9.05 | 26 % | 10.43 |

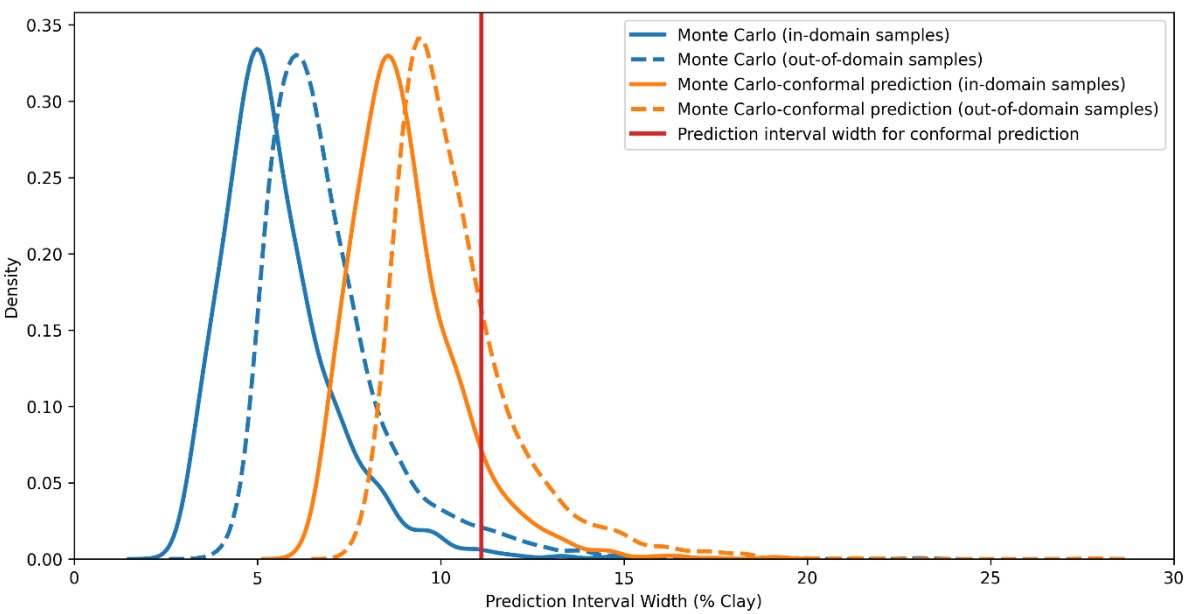

**Figure 4: Distribution of prediction interval width of Monte Carlo dropout, conformal prediction, and Monte Carlo-conformal prediction for in-domain and out-of-domain samples.**

On the other hand, both CP and MC-CP were able to achieve a coverage of 91 % (Fig. 3; Table 4), from the expected coverage of 90 %. This implied that 91 % of the prediction interval contained the true observed clay content, making the prediction interval reliable. However, the MPIWs of CP and MC-CP were higher than that of MC, indicating a trade-off between narrower intervals and coverage. The MPIW of CP (11.11 %) was the largest of the three methods, twice of MC dropout (5.56 %) (Fig. 4; Table 4), making it overly conservative. Additionally, the interval of CP was constant, which prohibited CP from addressing

the different uncertainties as "guaranteed coverage" was the main objective of this method. In other words, CP generated wide prediction intervals that were unnecessary.

The MC-CP method achieved a balance between MC dropout and CP, which produced an MPIW between MC dropout and CP while still reaching the expected coverage. The MPIW of MC-CP was 9.05 %, which was 1.6 times the MPIW of MC (Table 4) but achieved a 91 % coverage from the expected coverage of 90 %. Additionally, MC-CP retained the ability to address the uncertainty of out-of-domain samples, as the MPIW for out-of-domain samples (10.43 %) was larger than the MPIW for in-domain samples (9.05 %) (Fig. 4; Table 4). Hence, MC-CP is an adequate compromise among (1) coverage of observed values, (2) addressing out-of-domain uncertainty, and (3) reasonably sized MPIW.

However, when facing out-of-domain samples, MC-CP achieved only 26 % coverage at the 90 % prediction interval. The MPIW for out-of-domain samples (10.43 %) was 1.38 higher than that for in-domain samples (9.05 %), representing a 15 % increase in the width. The difference was insufficient to fully account for out-of-domain uncertainty, leading to the low coverage. Similarly, Liu et al. (2021) found that Bayesian neural networks and MC dropout were unable to assign high uncertainty to out-of-domain samples, indicating overconfidence in predicting unknown data. Zadorozhny et al. (2021) also highlighted the tendency of neural networks to overgeneralise from training data when predicting out-of-domain samples, potentially leading to overconfidence. When out-of-domain sample inputs closely resemble in-domain sample inputs, MC dropout may assign similar confidence levels to out-of-domain samples, failing to capture the true uncertainty. The MIR spectra of clayey soils were not as distinct from those of sandy soils as the spectra of high SOC soils were from low SOC soils (Ng et al., 2022; Zhang et al., 2022). For example, peaks at 2930-2850 $cm^{-1}$ serve as a distinction between mineral soils and organic soils (Tinti et al., 2015; Ng et al., 2022). Thus, the difference between the MPIW of in-domain and out-of-domain samples was not as significant as in the study of Padarian et al. (2022), in which 20% SOC was used as the separation between in-domain and out-of-domain samples.

### 3.3 Limitations and future applications

The MC-CP method was able to quantify the uncertainty and generate prediction intervals with sufficient coverage of true values. However, one obvious difference between MC-CP and MC is that MC-CP requires calibration samples to establish nonconformity scores. However, only a small number of calibration samples were required compared to the training samples, which can be easily achieved by dividing a portion of the training sample. For instance, the size of the calibration set in the MC-CP regression example presented by Bethell et al. (2024) was only 2 % of the testing samples. Future studies could explore determining the optimal size of calibration sets. Another potential enhancement of CP is the use of clustering to assign different PIWs to distinct groups. For example, Malone et al. (2011) applied fuzzy clustering prior to implementing UNEEC, which is similar to CP in calculation. With this means, the out-of-domain samples can be addressed by introducing an extragrade cluster. Additionally, asymmetric PIWs, where the upper and lower bounds differ in width, could be explored in the MC-CP framework to account for potential imbalance in prediction errors.

While CP is model-agnostic, MC dropout is restricted to deep neural networks since it requires the inclusion of dropout layers in the model architecture (Gal and Ghahramani, 2016). Hence, MC-CP is also model-specific and can only be used on deep neural networks. As neural networks have been widely applied in soil spectroscopy, with several studies reporting accurate prediction results (Ng et al., 2019; Padarian et al., 2019; Javadi et al., 2021). MC-CP offers a reliable method for generating prediction uncertainty without the computational burden associated with the Bayesian approach. Future investigations should compare these uncertainty quantification methods in terms of computational efficiency and MPIW.

Schmidinger and Heuvelink (2023) raised the issue that PICP ignores the one-sided bias in prediction, in which 90 % of the interval covers the observed value, but the probability outside the boundaries is asymmetrically distributed. Other parameters, such as quantile coverage probability and probability integral transform, are thus needed to evaluate the uncertainty

quantification in the future. In the present study, out-of-domain samples exhibited higher clay content than in-domain samples, with predicted values tending to be lower than the real value (Fig. 1). Consequently, observed values were more frequently above the 95 % quantile of prediction distribution, as illustrated by the out-of-domain example in Fig. 2. This one-sided bias arises from the separation of out-of-domain samples.

The efficiency of allows MC-CP to be applied to large models such as soil inference systems (McBratney et al., 2002), in which multiple pedotransfer functions were coupled together to predict complicated soil properties using basic soil properties that can be assessed from soil spectroscopy. Adding uncertainty analysis into model evaluation will increase the practicality of models and bring them one step closer to real-world applications.

## 4 Conclusions

The study aimed to assess the uncertainty in predicting clay content using convolutional neural networks through three uncertainty quantification techniques: Monte Carlo (MC) dropout, conformal prediction (CP), and MC-CP. The mid-infrared (MIR) spectra from the KSSL database were divided into two categories:

- In-domain samples
- Out-of-domain samples: This division tested the model's ability to handle samples that differ significantly from the training data.

The following methods were compared:

- MC Dropout:
  - Produced the lowest Prediction Interval Coverage Probability (PICP).
  - Generated the narrowest Mean Prediction Interval Width (MPIW), indicating overconfidence in predictions.
- Conformal Prediction (CP):
  - Achieved the ideal PICP but had a fixed and the largest MPIW among the methods.
- MC-CP:
  - Balanced the strengths of the other methods, achieving 91 % PICP (for a 90 % expected PICP) with a moderate MPIW.

The advantages of MC-CP are:

- Provides a balance between MC Dropout and CP.
- Exhibits:
  1. High coverage probability of true values.
  2. Variable prediction intervals that adapt to out-of-domain samples.
  3. Moderate MPIW for balanced uncertainty representation.

The main implications are:

- MC-CP demonstrates the potential for quantifying uncertainty in DL models for soil property prediction.
- The method allows for computationally efficient uncertainty quantification and producing prediction intervals that reliably cover the true values.

Future directions:

- Integration of MC-CP into large-scale prediction systems, such as soil inference models, to enhance prediction accuracy and support decision-making in real-world applications.

**Acknowledgements**

The authors acknowledge the staff at the National Soil Survey Center Kellogg Soil Survey Laboratory (Lincoln, NE) who have collected and analysed the soil samples in the KSSL dataset. We would like to thank the authors of Bethell et al. (2024), especially Daniel Bethell, for providing the original code of MC-CP. The study acknowledges funding from National Soil Carbon Innovation Challenge – Development and Demonstration Round 2 grant: An integrated schema for soil carbon stock estimation and crediting.

**Data and code availability**

The data used in this study are owned and managed by the National Soil Survey Center Kellogg Soil Survey Laboratory (KSSL). Interested parties should contact KSSL directly to request access, in accordance with their data-sharing policies. The code used to perform MC-CP in this study is available via a GitHub repository (https://doi.org/10.5281/zenodo.15401499, Huang, 2025, and https://github.com/LloydYCHuang/Soil-MC-CP).

**Author contributions**

YCH: Conceptualization, Data curation, Formal analysis, Investigation, Visualization, Writing – original draft preparation; JP: Conceptualization, Supervision, Writing – review & editing; BM: Supervision, Writing – review & editing; ABM: Supervision, Writing – review & editing.

**Competing interests**

The authors declare that they have no conflict of interest.

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
