# Peer review of "Using Monte Carlo conformal prediction to evaluate the uncertainty of deep learning soil spectral models"

_EGUsphere, 2024_

## Referee Comment (RC1)

**General**

The authors introduce "Monte Carlo conformal prediction" (MC-CP), a combination of Monte Carlo dropout and conformal prediction, for the uncertainty quantification of soil spectroscopy predictions with deep learning models. They demonstrate its merits compared to pure Monte Carlo dropout or conformal prediction approaches for in-domain predictions. Furthermore, the paper is logically structured and easy to follow. However, the introduction describes several concepts incompletely. Moreover, I do not share their conclusions regarding the "out-of-domain" predictions, as they claim that MC-CP can address uncertainty for out-of-domain data, even though all they show is that the prediction intervals are slightly wider for out-of-domain data, without providing information on the (presumably poor) coverage. These key issues need to be addressed prior to publication. See below specific comments:

**# Comment 1; L. 7**

The abstract should mention that Monte Carlo dropout is a method for neural networks (i.e., deep learning). Currently, it mentions "machine learning" in L. 7, which could appear as if the method is model-agnostic for (any) machine learning model. Either, machine learning could be replaced with deep learning, or it could be somewhere else explicitly mentioned that it is a method for deep learning.

**# Comment 2; L. 12**

The abstract should mention that Monte Carlo dropout is a method for neural networks (i.e., deep learning). Currently, it mentions "machine learning" in L. 7, which could appear as if the method is model-agnostic for (any) machine learning model. Either machine learning could be replaced with deep learning, or it could be explicitly mentioned somewhere else that it is a method for deep learning.

**Literature**

Bethell, D., Gerasimou, S., & Calinescu, R. (2024). Robust uncertainty quantification using conformalised Monte Carlo prediction. In Proceedings of the AAAI Conference on Artificial Intelligence (Vol. 38, No. 19, pp. 20939-20948).

Romano, Y., Patterson, E., & Candes, E. (2019). Conformalized quantile regression. Advances in neural information processing systems, 32.

**# Comment 3; L. 20 -21**

I do not share the opinion that MC-CP "effectively address[ed] the higher uncertainty in out-of-domain samples" given the results presented in this paper but my discussion on that can be found in Comment 19.

**# Comment 4; L. 23- 26**

I am convinced of the merits of MC-CP for deep learning in soil spectroscopy but the wording is exaggerated here. Neither "breakthrough" nor "revolutionizing" are appropriate terms here because the authors did not invent this method but demonstrated its advantage compared to their vanilla version for "in-domain data".

**Comment 5; L. 28-29**

*In the recent developments of soil science, machine learning has been widely used, such as soil spectroscopy, proximal sensing, carbon stock modelling, and digital soil mapping (Padarian et al., 2020; Minasny et al., 2024).* Wording around "such as" sounds slightly off, and I propose adding "in applications": "[..] widely used in applications such as soil spectroscopy, proximal sensing […]"

**Comment 6; L. 50 – 67**

The concept of aleatoric and epistemic uncertainty is quite vague and may even be incorrectly applied here because of the sentence in L. 54: "Epistemic uncertainty is the main topic in this study."

In the following, I use the definitions of Valdenegro-Toro & Mori (2022): "*There are two kinds of uncertainty […]: aleatoric or data uncertainty, and epistemic or model uncertainty. These uncertainties are usually combined and predicted as a single value, called predictive uncertainty […].*" Hence, the interest of the study is to find an uncertainty method (e.g. MC-CP) which succeeds in quantifying the combined predictive uncertainty.

Of course, epistemic uncertainty becomes especially relevant for the "out-of-domain data" predictions, as here epistemic uncertainty is very high. Hence, it is relevant that the uncertainty quantification model can also account for the epistemic uncertainty that is associated with the domain shift. Maybe this is what the authors intended to refer to but it is a far stretch to what is written and nowhere explicitly mentioned. Currently, it appears as if the authors confuse the predictive uncertainty with the epistemic uncertainty.

**Literature**

Valdenegro-Toro, M., & Mori, D. S. (2022, June). A deeper look into aleatoric and epistemic uncertainty disentanglement. In 2022 IEEE/CVF Conference on Computer Vision and Pattern Recognition Workshops (CVPRW) (pp. 1508-1516). IEEE.

**Comment 7; L. 57 – 58:**

Strictly speaking, this is inaccurate. An ideal uncertainty quantification method should have ideal coverage of the quantiles. Imagine a 90% prediction interval, which consists of a 5% and 95% quantile. So, an ideal uncertainty quantification method should have a 5% and 95% coverage of the quantiles. If the 5% quantile is covered by 2% of the test samples, and the 95% quantile by 92% of the test samples, the PICP would be still 90%, even though the uncertainty was wrongly quantified. See for example the short paper on PICP from Pinson & Tatsu (2014) or figure below.

[Figure]

[Figure]

**Literature**

Pinson, P., & Tastu, J. (2014). Discussion of "Prediction intervals for short-term wind farm generation forecasts" and "Combined nonparametric prediction intervals for wind power generation". IEEE Transactions on Sustainable Energy, 5(3), 1019-1020.

**Comment 8; L. 61 – 66**

It is hard to understand why the authors mention only bootstrapping in this context. It is correct, that bootstrapping is used occasionally in soil science to quantify uncertainties. However, this is a common methodological error because bootstrapping only creates "confidence intervals" not "prediction intervals" i.e., pure bootstrapping was never intended to be used to quantify predictive uncertainties. On the other hand, the authors leave out the most commonly used method in soil: quantile regression (e.g., quantile regression forest, XGBoost with quantile loss function etc.) or even conformalized quantile regression. Quantile regression is not (yet?) well implemented for deep learning, which is why MC-CP becomes relevant but the introduction feels incomplete in the context of soil. More so, because conformalized quantile regression was recently introduced in soil by Kakhani et al. (2024), which is in its logic very related to MC-CP.

**Literature**

Kakhani, N., Alamdar, S., Kebonye, N. M., Amani, M., & Scholten, T. (2024). Uncertainty quantification of soil organic carbon estimation from remote sensing data with conformal prediction. Remote Sensing, 16(3), 438.

**Comment 9; L. 76 – 86**

The concept of aleatoric and epistemic uncertainty may be applied to this section.

**Comment 10; L. 93 – L. 95**

One may argue that quantile regression could do so too, which is why it needs to be discussed somewhere earlier.

**Comment 11; L. 96**

More appropriate would be to replace "we applied a strategy to increase the PICP of MC dropout" with "we applied a strategy to improve the PICP coverage of MC dropout", unless MC generally leads to too narrow prediction intervals and not suboptimal coverage.

**Comment 12; L. 118 – 120:**

During my first read, I was wondering how the testing and validation was done. It is mentioned in L. 203 but for better readability it could be already defined here. I agree with it either way.

**Comment 13; L. 123**

It may be considered to use "it has been proven" instead of "it has been proved".

**# Comment 14; L. 134:**

Sounds slightly off. One suggestion to highlight that the predictive distribution is inferred from 100 trained CNN models with dropout layers: "In practice, a CNN model with dropout layers was trained 100 times to generate a predictive distribution".

**# Comment 15; L. 135-136:**

The 90 % prediction interval is not the difference between the 5th and 95th quantile, but the prediction interval itself is the interval defined by those two quantiles. The difference is the "(mean) prediction interval width". In Eq. 1 it is shown correctly, meaning it is just an issue of terminology.

**# Comment 16; Eq. 1/ L.137:**

$C$ is the 90% prediction interval, which is relatively logical given the previous sentence. Nonetheless, it could be defined in the text. Also, if it is called $C_{90}$, it would follow the scheme of $\hat{q}_5$ and $\hat{q}_{95}$.

**# Comment 17; L. 222:**

Wording could be considered: it is correct that it is expectable that the model fails to do proper out-of-domain predictions, but the performance is not only "poor" but completely unusable since the $R^2$ = -6.64. The word "poor" indicates to me a model with an $R^2$ around or slightly above 0.

**# Comment 18; L. 250/Fig. 3:**

It seems a bit incoherent that the plot covers the $0 - 90\%$ range instead of $0 - 100\%$ range.

**# Comment 19; Out-of-domain results and discussion:**

I do not fully agree with the discussion on the out-of-domain results for the following reasons:

First of all, it is not clear why the authors do not show the PICP for the out-of-domain predictions, presumably because it has very poor coverage, given that the MPIW is only marginally larger even though the model is extremely bad for out-of-domain predictions. A much larger uncertainty (i.e., MPIW) should be expected here.

Instead, the authors focus on the fact that the MPIW is slightly larger for out-of-domain predictions with MC-CP and MC compared to in-domain predictions. It correctly shows that the model is somewhat aware that it is less certain for out-of-domain data. The authors see this as a reason to conclude that MC-CP is able to address the uncertainty of out-of-domain samples (L. 278-281). However, the results do not really support this claim because the MPIW alone is uninformative without information on the coverage. I highly assume that the MPIWs are still not wide enough. For extensive conclusions, the authors should include the coverage for the out-of-domain predictions. Hence, the coverage of the out-of-domain data needs to be included as well!

A second proceeding problem may occur if the PICP is used for evaluating out-of-domain samples, and it is strongly associated with the previous comment 7.

It can be expected that the observed values will be much more likely to be above the 95% quantile (as shown in the right example of Fig. 2!) and less often below the 5% quantile because the observed out-of-domain values have higher clay values than the model is trained on. Hence, evaluating the quantiles would make much more sense than using the PICP. This is the more standard practice and has been addressed in the context of soil too (Schmidinger & Heuvelink, 2023).

**Literature**

Schmidinger, J., & Heuvelink, G. B. (2023). Validation of uncertainty predictions in digital soil mapping. Geoderma, 437, 116585.

**# Comment 20; L. 255:**

MPIW instead of PIW.

**# Comment 21; L. 344:**

"The authors benefited from the shared code of Daniel Bethell. The manuscript would become much more impactful for the soil community if the authors shared their code as well. This would increase the usability of the MC-CP method. Given that some KSSL data has been published in OSSL, it would also be easy to reproduce the study."

---

## Referee Comment (RC2)

**General Revision Summary**

The study presents a well-executed analysis of uncertainty quantification in deep learning soil spectral models, specifically through the use of Monte Carlo-Conformal Prediction (MC-CP). The paper makes a strong contribution to the field by addressing a crucial gap in soil spectroscopy—reliable uncertainty quantification. The methodological approach is well-documented, and the comparison between MC dropout, Conformal Prediction (CP), and MC-CP is insightful and thorough.

**Strengths of the Paper**

- Novel Contribution: The paper introduces MC-CP as an method for uncertainty quantification in deep learning soil spectral models. The demonstration of MC-CP's ability to balance expected coverage, computational efficiency, and adaptability to out-of-domain samples is a significant advancement.
- Well-Designed Comparison: The comparison between MC dropout, CP, and MC-CP is informative and shows the trade-offs between these methods.
- Strong Methodological Foundation: The study follows a solid methodological framework.
- Practical Relevance: The application of the proposed method to real-world soil spectral data enhances the practical impact of the study

**General Improvements**

Terminology and Consistency (Machine Learning vs. Deep Learning)

- The abstract and introduction interchangeably refer to Machine Learning (ML) and Deep Learning (DL). However, the methodology and model used are specifically deep learning-based. Ensure consistency in terminology and explicitly state where ML is a broader category and where DL is specifically applied.
- Incorporate a broader range of examples in the Introduction for Monte Carlo (MC) Dropout and Conformal Prediction (CP), as the current section focuses too narrowly on just two detailed examples.
- There is a lack of clear structure (detailed further in the comments). There are many redundant repetitions, and subordinate clauses with very general information are interspersed throughout, often repeating details that were already mentioned earlier.
- Ensure there is a space between the number and the percentage symbol for proper formatting.

**Detailed Comments**

| Comment No | Lines | *Original* | Review |
|---|---|---|---|
| 1 | 7-10 | *"While machine learning has made remarkable strides in predicting various physiochemical properties of soils using spectroscopy, predictions devoid of quantified uncertainty offer limited utility in guiding critical decisions. However,* | The sentence effectively explains that predictions without uncertainty are not useful for decision-making and that uncertainty quantification is rarely used due to limitations. However, the logical connection between these points could be clearer to improve readability and coherence. |

| | | *uncertainty quantification remains underutilised in the reporting of soil spectral models, with existing methods facing significant limitations."* | |
|---|---|---|---|
| 2 | 10 | *" These approaches are either computationally demanding…."* | It is not entirely clear whether this refers to the existing methods mentioned in the previous sentence or to something else, as methods and approaches are not necessarily the same. |
| 3 | 11-23 | - | The structure is confusing in the sense that your method is mentioned without prior explanation, followed by the introduction of two established methods for comparison. Additionally, while introducing these methods, you already include some results. To improve clarity, consider restructuring the section by clearly separating the description of methods, the comparison, and then presenting the results. |
| 4 | 24-26 | *"This breakthrough enhances the real-world applicability of soil spectral models and represents a significant advancement in the field of soil science. [….] further revolutionising decision-making and risk assessment in soil science."* | Shorten this section to two sentences, as the usefulness is stated twice. Avoid redundant explanations to improve clarity. |
| 5 | 29 | *"[…](Padarian et al., 2020; Minasny et al., 2024). These studies are characterised 30 by the use of large soil datasets and require an efficient way of extracting information to predict target attributes."* | The reference is incorrect, as these studies do not discuss what you describe in the following sentence. |
| 6 | 41-46 | | There are repetitions in the sentences without adding new content. Shorten them for conciseness. |
| 7 | 43 | *"Despite the significant success of machine learning in predicting soil properties, uncertainty quantification of the prediction remained an underexplored area in soil spectroscopy, and only a few studies have tried to include uncertainty in the model evaluation."* | A reference is needed for the studies mentioned. |

| 8 | 50-54 | | I don't see the relevance of explaining the difference between the two types of uncertainty here, as it does not appear to be a topic in the methods section or the discussion. |
|---|---|---|---|
| 9 | 61-66 | | To my knowledge, bootstrapping is typically used for confidence intervals, not for prediction intervals like MC and CP. Additionally, different methods of quantile regression and Gaussian methods are missing, which would help provide a more complete introduction. |
| 10 | 68-72 | | Specify that MC is specifically used for deep learning to avoid ambiguity. |
| 11 | 96-103 | *"In this study, we applied a strategy to increase the PICP of MC dropout while maintaining its advantages in characterising out-of-domain uncertainty. Monte Carlo-Conformal Prediction (MC-CP) was introduced by Bethell et al. (2024). MC-CP integrates the strengths of both MC dropout and CP."* | Clarify that MC-CP is the strategy. Again, avoid repetition to improve clarity and conciseness. |
| 12 | 113-115 | | Please specify how many of the removed samples were due to SOC and how many were excluded because of extreme values. |
| 13 | 116 | | Clarify why the threshold of 40% clay content was chosen and provide justification for this choice. |
| 14 | 119 | | If you are already describing your training and test scheme here, also include the ratio of the splitting mentioned in L203 for consistency and completeness. |
| 15 | Chapter 2.2, 2.3, 2.4 | | For better structure, I suggest organizing the section as follows: 2.2 Methods, with subsections 2.2.1 Monte Carlo Dropout (MC dropout)**,** 2.2.2 Conformal Prediction (CP)**,** and 2.2.3 Monte Carlo-Conformal Prediction (MC-CP)**.** |
| 16 | 125 | | Missing abbreviation: MC dropout |
| 17 | 128 | *"In each dropout layer, a certain portion of the neurons is randomly deactivated (weights set to zero) during both training and testing."* | As far as I know, and as stated in the paper by Gal and Ghahramani (2016), neurons are only deactivated during training. While validation can be involved, a specific reason is needed for doing so. Please verify what is happening in your specific use case. |
| 18 | 137 | | Check the Mathematical notation and terminology of the journal: https://publications.copernicus.org/for_authors/manuscript_preparation.html#math. I recommend centering the equations for better readability. Additionally, equations should be treated as nouns within the text. So here I would change it to the following: The 90% prediction interval […] of the predictions (Eq. 1): Formula. (Eq. 1) |
| 19 | 137 | | When using a formula, ensure that every abbreviation is defined either before or in the sentence following it. In this case, $C_{MC}$ and $X_i$ are missing definitions. |

| 20 | 150 | *Table 1* | Stay consistent in using **X** or **X**$_i$ throughout the table to maintain clarity and uniformity. |
|----|-----|-----------|----------------------------------------------------------------------------------------------------------|
| 21 | 161 | | See comment No. 18 |
| 22 | 170 | | Stay consistent in the writing of Monte Carlo-conformal prediction**.** Since it is based on Bethell et al. (2024)**,** I recommend following their terminology and formatting. |
| 23 | 179 | | See comment No. 18 |
| 24 | 184 | | See comment No. 18 |
| 25 | 208-209 | | See comment No. 18 and a reference is missing for the Eq. 5 and 6. |
| 26 | 210-214 | | See comment No. 18 a space is missing in Eq. 7 between the fraction and "count". |
| 27 | 223 | | I would rephrase it as follows, omitting the word "poor": "A negative R-squared value indicates that the model performs worse than simply using the mean prediction." |
| 28 | 224-225 | | Connect the two sentences for example as following: "Such results for out-of-domain samples were expected, as the model did not have any knowledge of soils with clay content larger than 40%, leading most out-of-domain predictions to fall under 40% clay." |
| 29 | 238 | *"When the evaluation of uncertainty is optimal, the expected coverage of a $p\%$ prediction interval is $p\%$ (dotted line in Fig. 3)"* | What do you mean by "evaluation of uncertainty"? Please clarify or provide a more precise definition. |
| 30 | 255 | | MPIW instead of PIW |
| 31 | 263 | *Table 4* | The PICP value for out-of-domain samples is missing and should be included for completeness. |
| 32 | 276-281 | | I do not agree with the strong wording that MC-CP effectively addresses the out-of-domain issue, as the difference in MPIW between in-domain and out-of-domain samples is not significant. |
| 33 | 299-304 | | This part should be discussed directly in the uncertainty section rather than in the limitations and future applications section for better coherence. |
| 34 | 312 | | Specify the exact deep learning model used. |
| 35 | 329 | | The wording should be revised—for an optimal trade-off, the results need to be more significant. |

---

## Author Comment (AC1)

**Referee #1:**

General

The authors introduce "Monte Carlo conformal prediction" (MC-CP), a combination of Monte Carlo dropout and conformal prediction, for the uncertainty quantification of soil spectroscopy predictions with deep learning models. They demonstrate its merits compared to pure Monte Carlo dropout or conformal prediction approaches for in-domain predictions. Furthermore, the paper is logically structured and easy to follow. However, the introduction describes several concepts incompletely. Moreover, I do not share their conclusions regarding the "out-of-domain" predictions, as they claim that MC-CP can address uncertainty for out-of-domain data, even though all they show is that the prediction intervals are slightly wider for out-of-domain data, without providing information on the (presumably poor) coverage. These key issues need to be addressed prior to publication. See below specific comments:

**Reply:** Thank you for your comments. We will address them in the following replies.

**Comment 1; L. 7**

The abstract should mention that Monte Carlo dropout is a method for neural networks (i.e., deep learning). Currently, it mentions "machine learning" in L. 7, which could appear as if the method is model-agnostic for (any) machine learning model. Either, machine learning could be replaced with deep learning, or it could be somewhere else explicitly mentioned that it is a method for deep learning.

**Reply:** We agree. We change sentence in the abstract to mention that MC-CP is a method for deep learning.

"This study introduces an innovative application of Monte Carlo conformal prediction (MC-CP) to quantify uncertainty in deep learning models for predicting clay content from mid-infrared spectroscopy."

**Comment 2; L. 12 EDIT**

The more common name for this method is not "Monte Carlo conformal prediction" but "conformalised Monte Carlo prediction" (Bethell et al. 2024) following the predecessor "conformalized quantile regression" (Romano et al. 2019). However, I do not consider it a mistake because both variants exist.

Literature

Bethell, D., Gerasimou, S., & Calinescu, R. (2024). Robust uncertainty quantification using conformalised Monte Carlo prediction. In Proceedings of the AAAI Conference on Artificial Intelligence (Vol. 38, No. 19, pp. 20939-20948).

Romano, Y., Patterson, E., & Candes, E. (2019). Conformalized quantile regression. Advances in neural information processing systems, 32.

**Reply:** Thank you for your understanding. We decided to keep this name as it can indicate the combination of two methods, and this name also appears in Bethell et al. (2024).

**Comment 3; L. 20 -21**

I do not share the opinion that MC-CP "effectively address[ed] the higher uncertainty in out-of-domain samples" given the results presented in this paper but my discussion on that can be found in Comment 19.

**Reply:** Thank you. We remove the strong wording "effectively" and address this issue together in Comment 19.

**Comment 4; L. 23- 26**

I am convinced of the merits of MC-CP for deep learning in soil spectroscopy but the wording is exaggerated here. Neither "breakthrough" nor "revolutionizing" are appropriate terms here

because the authors did not invent this method but demonstrated its advantage compared to their vanilla version for "in-domain data".

**Reply:** We revise the wordings mentioned above.

"The success of MC-CP enhances the real-world applicability of soil spectral models, paving the way for their integration into large-scale machine-learning models, such as soil inference systems, and further transforming decision-making and risk assessment in soil science."

**Comment 5; L. 28-29**

In the recent developments of soil science, machine learning has been widely used, such as soil spectroscopy, proximal sensing, carbon stock modelling, and digital soil mapping (Padarian et al., 2020; Minasny et al., 2024). Wording around "such as" sounds slightly off, and I propose adding "in applications": "[..] widely used in applications such as soil spectroscopy, proximal sensing […]"

**Reply:** Thank you for the suggestion. We revised it accordingly.

"In the recent developments of soil science, machine learning has been widely used in applications such as soil spectroscopy, proximal sensing, carbon stock modelling, and digital soil mapping (Ng et al., 2019; Wadoux et al., 2020)."

**Comment 6; L. 50 – 67**

The concept of aleatoric and epistemic uncertainty is quite vague and may even be incorrectly applied here because of the sentence in L. 54: "Epistemic uncertainty is the main topic in this study."

In the following, I use the definitions of Valdenegro-Toro & Mori (2022): "There are two kinds of uncertainty […]: aleatoric or data uncertainty, and epistemic or model uncertainty. These uncertainties are usually combined and predicted as a single value, called predictive uncertainty

[…]." Hence, the interest of the study is to find an uncertainty method (e.g. MC-CP) which succeeds in quantifying the combined predictive uncertainty.

Of course, epistemic uncertainty becomes especially relevant for the "out-of-domain data" predictions, as here epistemic uncertainty is very high. Hence, it is relevant that the uncertainty quantification model can also account for the epistemic uncertainty that is associated with the domain shift. Maybe this is what the authors intended to refer to but it is a far stretch to what is written and nowhere explicitly mentioned. Currently, it appears as if the authors confuse the predictive uncertainty with the epistemic uncertainty.

Literature

Valdenegro-Toro, M., & Mori, D. S. (2022, June). A deeper look into aleatoric and epistemic uncertainty disentanglement. In 2022 IEEE/CVF Conference on Computer Vision and Pattern Recognition Workshops (CVPRW) (pp. 1508-1516). IEEE.

**Reply:** Thank you for the suggestions. We understand that the interest of this study is on the combined predictive uncertainty. We decided to remove this part of introduction following the suggestion from referee #2 since our method and discussion did not further elaborate on this topic.

**Comment 7; L. 57 – 58:**

Strictly speaking, this is inaccurate. An ideal uncertainty quantification method should have ideal coverage of the quantiles. Imagine a 90% prediction interval, which consists of a 5% and 95% quantile. So, an ideal uncertainty quantification method should have a 5% and 95% coverage of the quantiles. If the 5% quantile is covered by 2% of the test samples, and the 95% quantile by 92% of the test samples, the PICP would be still 90%, even though the uncertainty was wrongly quantified. See for example the short paper on PICP from Pinson & Tatsu (2014) or figure below.

[Figure]

[Figure]

Literature

Pinson, P., & Tastu, J. (2014). Discussion of "Prediction intervals for short-term wind farm generation forecasts" and "Combined nonparametric prediction intervals for wind power generation". IEEE Transactions on Sustainable Energy, 5(3), 1019-1020.

**Reply:** Thank you. We agree that the evaluation with PICP could be potentially biased, and evaluating the quantile is a way to address this issue. Methods suggested by Schmidinger and Heuvelink (2023), namely quantile coverage probability and probability integral transform, could address these issues better than PICP.

However, regarding the method used in this study, CP only provides a 90% interval instead of a predictive distribution. Predictive CDFs that are required for probability integral transform cannot be directly calculated. Since MCCP is just MC with extended upper and lower bounds, its prediction distribution is also derived from the original MC.

Indeed, quantile of CP can potentially be calculated, but it won't be derived from a predictive distribution. We acknowledge the issue about one-sided bias and have a section in the 3.3 Limitations and future applications to discuss this issue.

"Schmidinger and Heuvelink (2023) raised the issue that PICP ignores the one-sided bias in prediction, in which 90 % of the interval covers the observed value, but the probability outside the boundaries is asymmetrically distributed. Other parameters, such as quantile coverage probability and probability integral transform, are thus needed to evaluate the uncertainty quantification in the future."

**Comment 8; L. 61 – 66**

It is hard to understand why the authors mention only bootstrapping in this context. It is correct, that bootstrapping is used occasionally in soil science to quantify uncertainties. However, this is a common methodological error because bootstrapping only creates "confidence intervals" not "prediction intervals" i.e., pure bootstrapping was never intended to be used to quantify predictive uncertainties. On the other hand, the authors leave out the most commonly used method in soil: quantile regression (e.g., quantile regression forest, XGBoost with quantile loss function etc.) or even conformalized quantile regression. Quantile regression is not (yet?) well implemented for deep learning, which is why MC-CP becomes relevant but the introduction feels incomplete in the context of soil. More so, because conformalized quantile regression was recently introduced in soil by Kakhani et al. (2024), which is in its logic very related to MC-CP.

Literature

Kakhani, N., Alamdar, S., Kebonye, N. M., Amani, M., & Scholten, T. (2024). Uncertainty quantification of soil organic carbon estimation from remote sensing data with conformal prediction. Remote Sensing, 16(3), 438.

Reply: Thank you for the suggestions. We revised the content to address these issues:

1. We added contents to indicate that bootstrapping is only used for confidence intervals instead of prediction intervals.

   "In addition, bootstrapping primarily addresses the model uncertainty and it derives confidence intervals rather than prediction intervals (Heuvelink, 2014; Wadoux, 2019)."

2. We added introduction about quantile regression and Bayesian CNNs to provide a more complete introduction.

   "The diverse nature of models enabled the development of different methods. For example, quantile regression (QR) uses a set of regression models to estimate the quantile of target

variables, and the prediction interval can later be defined by the upper and lower quantiles (Kasraei et al., 2021). Additionally, quantile regression forests (QRF) and quantile regression neural networks (QRNN) are also extensions of quantile regression that apply similar principles to generate prediction intervals (Schmidinger and Heuvelink, 2023). Heuvelink et al. (2021) utilised QRF to predict the SOC for soils in Argentina with quantified uncertainty, and the 0.05 and 0.95 quantiles were used to generate the 90 % prediction interval. On the other hand, Omondiagbe et al. (2024) compared bootstrapped PLS, generalised additive models (GAM), and Bayesian CNNs for their ability to quantify uncertainty. They found that GAM and Bayesian CNN outperformed bootstrapped PLS by having PICP close to the ideal 90% value. Moreover, the MPIW of Bayesian CNN is mostly lower than that of GAM models, suggesting a more accurate estimation of uncertainty (Omondiagbe et al., 2024). However, Bayesian neural networks are more intensive in computation compared to standard CNNs (Bethell et al., 2024; Omondiagbe et al., 2024)."

References:

Heuvelink, G. B.: Uncertainty quantification of GlobalSoilMap products, GlobalSoilMap: Basis of the global spatial soil information system, 335-340, 2014.

Heuvelink, G. B. M., Angelini, M. E., Poggio, L., Bai, Z., Batjes, N. H., van den Bosch, R., Bossio, D., Estella, S., Lehmann, J., Olmedo, G. F., and Sanderman, J.: Machine learning in space and time for modelling soil organic carbon change, Eur. J. Soil Sci., 72(4), 1607-1623, https://doi.org/10.1111/ejss.12998, 2021.

Kasraei, B., Heung, B., Saurette, D. D., Schmidt, M. G., Bulmer, C. E., and Bethel, W.: Quantile regression as a generic approach for estimating uncertainty of digital soil maps produced from machine-learning, Environmental Modelling & Software, 144, 105139, https://doi.org/10.1016/j.envsoft.2021.105139, 2021.

Omondiagbe, O. P., Roudier, P., Lilburne, L., Ma, Y., and McNeill, S.: Quantifying uncertainty in the prediction of soil properties using mid-infrared spectra, Geoderma, 448, 116954, https://doi.org/10.1016/j.geoderma.2024.116954, 2024.

Wadoux, A. M. J. C.: Using deep learning for multivariate mapping of soil with quantified uncertainty, Geoderma, 351, 59-70, https://doi.org/10.1016/j.geoderma.2019.05.012, 2019.

**Comment 9; L. 76 – 86**

The concept of aleatoric and epistemic uncertainty may be applied to this section.

**Reply:** We removed the introduction about aleatoric and epistemic uncertainty.

**Comment 10; L. 93 – L. 95**

One may argue that quantile regression could do so too, which is why it needs to be discussed somewhere earlier.

**Reply:** We added introduction about quantile regression in Comment 8.

**Comment 11; L. 96**

More appropriate would be to replace "we applied a strategy to increase the PICP of MC dropout" with "we applied a strategy to improve the PICP coverage of MC dropout", unless MC generally leads to too narrow prediction intervals and not suboptimal coverage.

**Reply:** We revised it accordingly.

**Comment 12; L. 118 – 120:**

During my first read, I was wondering how the testing and validation was done. It is mentioned in L. 203 but for better readability it could be already defined here. I agree with it either way.

**Reply:** Thank you for the suggestion. We moved L203 here to indicate the data split during

modelling.

"The in-domain data were further randomly separated into 85 % training, 5 % validation, 5 % calibration for conformal prediction, and 5 % testing. Only the training and validation data were used in building the model."

**Comment 13; L. 123**

It may be considered to use "it has been proven" instead of "it has been proved".

**Reply:** We corrected it accordingly.

**Comment 14; L. 134:**

Sounds slightly off. One suggestion to highlight that the predictive distribution is inferred from 100 trained CNN models with dropout layers: "In practice, a CNN model with dropout layers was trained 100 times to generate a predictive distribution".

**Reply:** Thank you for the comment. Monte Carlo dropout method generates predictive distribution by performing numerous (in this case 100) forward passes with the same CNN model with dropout activated.

From Gal and Ghahramani (2016): *"Note that the dropout NN model itself is not changed. To estimate the predictive mean and predictive uncertainty we simply collect the results of stochastic forward passes through the model."*

From Bethell et al. (2024): *"Although dropout is typically used during training, MC dropout keeps this feature active during inference and performs several forward passes to devise a prediction distribution."*

We modify this sentence to clarify this process:

"In practice, a CNN model with dropout layers was trained and performed 100 forward passes with dropout layers activated to generate a predictive distribution (Bethell et al., 2024)."

References:

Bethell, D., Gerasimou, S., and Calinescu, R.: Robust Uncertainty Quantification Using Conformalised Monte Carlo Prediction, Proceedings of the AAAI Conference on Artificial Intelligence, 38(19), 20939-20948, https://doi.org/10.1609/aaai.v38i19.30084, 2024.

Gal, Y., and Ghahramani, Z.: Dropout as a Bayesian Approximation: Representing Model Uncertainty in Deep Learning, Proceedings of the 33rd International Conference on Machine Learning, 48, 1050-1059, https://proceedings.mlr.press/v48/gal16.html, 2016.

**Comment 15; L. 135-136:**

The 90 % prediction interval is not the difference between the 5th and 95th quantile, but the prediction interval itself is the interval defined by those two quantiles. The difference is the "(mean) prediction interval width". In Eq. 1 it is shown correctly, meaning it is just an issue of terminology.

**Reply:** Thank you for correcting this. We revise the content as follows:

"The 90 % prediction interval of MC dropout ($C_{MC,\,90}$) of each sample $i$ would be defined by the 5$^{th}$ quantile ($\hat{q}_5(X_i)$) and the 95$^{th}$ quantile ($\hat{q}_{95}(X_i)$) of the predictions (Eq. 1)"

**Comment 16; Eq. 1/ L.137:**

C is the 90% prediction interval, which is relatively logical given the previous sentence. Nonetheless, it could be defined in the text. Also, if it is called C90, it would follow the scheme of $\hat{q}5$ and $\hat{q}95$.

**Reply:** Thank you. We define that $C_{MC,\,90}$ indicates the 90 % prediction interval in the text.

"The 90 % prediction interval of MC dropout ($C_{MC,\,90}$) of each sample $i$ would be defined by the 5$^{th}$ quantile ($\hat{q}_5(X_i)$) and the 95$^{th}$ quantile ($\hat{q}_{95}(X_i)$) of the predictions (Eq. 1)"

**Comment 17; L. 222:**

Wording could be considered: it is correct that it is expectable that the model fails to do proper out-of-domain predictions, but the performance is not only "poor" but completely unusable since the R2 = -6.64. The word "poor" indicates to me a model with an R2 around or slightly above 0.

**Reply:** We agree. Combining the suggestion from referee #2, we modified the sentence as follows:

"For out-of-domain samples, a negative R-squared value indicates that the model performs worse than simply using the mean prediction"

**Comment 18; L. 250/Fig. 3:**

It seems a bit incoherent that the plot covers the 0 – 90% range instead of 0 – 100% range.

**Reply:** Fig. 3 is now as follows:

[Figure]

Figure 3: Prediction interval coverage probability (PICP) of in-domain and out-of-domain samples at different prediction intervals for Monte Carlo dropout, conformal prediction, and Monte Carlo-conformal prediction.

**Comment 19; Out-of-domain results and discussion:**

I do not fully agree with the discussion on the out-of-domain results for the following reasons: First of all, it is not clear why the authors do not show the PICP for the out-of-domain predictions, presumably because it has very poor coverage, given that the MPIW is only marginally larger even though the model is extremely bad for out-of-domain predictions. A much larger uncertainty (i.e., MPIW) should be expected here.

Instead, the authors focus on the fact that the MPIW is slightly larger for out-of-domain predictions with MC-CP and MC compared to in-domain predictions. It correctly shows that the model is somewhat aware that it is less certain for out-of-domain data. The authors see this as a reason to conclude that MC-CP is able to address the uncertainty of out-of-domain samples (L. 278-281). However, the results do not really support this claim because the MPIW alone is uninformative without information on the coverage. I highly assume that the MPIWs are still not wide enough. For extensive conclusions, the authors should include the coverage for the out-of-domain predictions. Hence, the coverage of the out-of-domain data needs to be included as well!

A second proceeding problem may occur if the PICP is used for evaluating out-of-domain samples, and it is strongly associated with the previous comment 7.

It can be expected that the observed values will be much more likely to be above the 95% quantile (as shown in the right example of Fig. 2!) and less often below the 5% quantile because the observed out-of-domain values have higher clay values than the model is trained on. Hence, evaluating the quantiles would make much more sense than using the PICP. This is the more standard practice and has been addressed in the context of soil too (Schmidinger & Heuvelink, 2023).

Literature

Schmidinger, J., & Heuvelink, G. B. (2023). Validation of uncertainty predictions in digital soil mapping. Geoderma, 437, 116585.

**Reply:** Thank you. We made necessary updates to address this comment.

1.  We agree that the PICP of out-of-domain samples are also important for the context. We
    updated Fig. 3 and Table 4 to include the PICP of out-of-domain samples:

[Figure]

Figure 3: Prediction interval coverage probability (PICP) of in-domain and out-of-domain
samples at different prediction intervals for Monte Carlo dropout, conformal prediction, and
Monte Carlo-conformal prediction.

Table 4: Results of uncertainty quantification by Monte Carlo dropout, conformal prediction,
and Monte Carlo-conformal prediction. PICP stands for prediction interval coverage
probability, and MPIW stands for mean prediction interval width.

| Method | 90 % PICP in-domain | MPIW (%) in-domain | 90 % PICP out-of-domain | MPIW (%) out-of-domain |
|---|---|---|---|---|
| Monte Carlo dropout | 74 % | 5.56 | 11 % | 6.90 |
| Conformal prediction | 91 % | 11.11 | 32 % | 11.11 |
| Monte Carlo-conformal prediction | 91 % | 9.05 | 26 % | 10.43 |

2. We acknowledge that the MPIW of out-of-domain samples was not dramatically different from the MPIW of in-domain samples. In our example, the MPIW of out-of-domain samples (10.43 %) was different from the MPIW of in-domain samples (9.05 %) by 1.38 %, which was about 15 % (1.38/9.05) increasement of width. We added discussion about the possible reasons behind such performance. In Liu et al. (2021), the authors found that Bayesian NN and MC dropout did not give high uncertainty to out-of-domain samples. Zadorozhny et al. (2021) discussed the problem that sometimes the NN over-generalise from training data to predict out-of-domain samples. That is, when the out-of-domain samples are similar to the in-domain samples, the algorithm wrongly assign high confidence to the prediction of out-of-domain samples. This could be the case in our study, as the spectra of clayey soils were not as distinct from those of sandy soils as the spectra of high SOC soils were from low SOC soils. The spectra in Ng et al. (2022) and Zhang et al. (2022) can support this argument as the high SOC data have significant peaks at 2930-2850 cm$^{-1}$ due to alkyl groups in SOM. Additionally, the difference of MIR spectra between mineral soils and organic soils could be bigger since this plot only includes soil samples with organic carbon less than 12%.

[Figure]

Figure from Ng et al. (2022)

[Figure]

Figure from Zhang et al. (2022)

We included the above-mentioned discussion about the possible reasons that MC-CP did not generate a wider MPIW for out-of-domain samples:

"However, when facing out-of-domain samples, MC-CP achieved only 26 % coverage at the 90 % prediction interval. The MPIW for out-of-domain samples (10.43 %) was 1.38 higher than that for in-domain samples (9.05 %), representing a 15 % increase in the width. The difference was insufficient to fully account for out-of-domain uncertainty, leading to the low coverage. Similarly, Liu et al. (2021) found that Bayesian neural networks and MC dropout were unable to assign high uncertainty to out-of-domain samples, indicating overconfidence in predicting unknown data. Zadorozhny et al. (2021) also highlighted the tendency of neural networks to overgeneralise from training data when predicting out-of-domain samples, potentially leading to overconfidence. When out-of-domain sample inputs closely resemble in-domain sample inputs, MC dropout may assign similar confidence levels to out-of-domain samples, failing to capture the true uncertainty. The MIR spectra of clayey soils were not as distinct from those of sandy soils as the spectra of high SOC soils

were from low SOC soils (Ng et al., 2022; Zhang et al., 2022). For example, peaks at 2930-2850 cm-1 serve as a distinction between mineral soils and organic soils (Tinti et al., 2015; Ng et al., 2022). Thus, the difference between the MPIW of in-domain and out-of-domain samples was not as significant as in the study of Padarian et al. (2022), in which 20% SOC was used as the separation between in-domain and out-of-domain samples."

3.  Yes, we agree that the out-of-domain samples are more likely to be above the 95% quantile. This is reasonable since the out-of-domain samples have clay contents way higher than the in-domain samples, and the model always underpredict the clay content. We have a short discussion about the one-sided bias of out-of-domain samples:

    "In the present study, out-of-domain samples exhibited higher clay content than in-domain samples, with predicted values tending to be lower than the real value (Fig. 1). Consequently, observed values were more frequently above the 95 % quantile of prediction distribution, as illustrated by the out-of-domain example in Fig. 2. This one-sided bias arises from the separation of out-of-domain samples."

References:

Liu, Y., Pagliardini, M., Chavdarova, T., and Stich, S. U.: The Peril of Popular Deep Learning Uncertainty Estimation Methods, Proceedings of the Bayesian Deep Learning workshop, NeurIPS 2021, https://doi.org/10.48550/arXiv.2112.05000, 2021.

Ng, W., Minasny, B., Jeon, S. H., and McBratney, A.: Mid-infrared spectroscopy for accurate measurement of an extensive set of soil properties for assessing soil functions, Soil Secur., 6, 100043, https://doi.org/10.1016/j.soisec.2022.100043, 2022.

Zadorozhny, K., Ulmer, D., and Cinà, G.: Failures of Uncertainty Estimation on Out-Of-Distribution Samples: Experimental Results from Medical Applications Lead to Theoretical

Insights, Proceedings of the ICML 2021 Workshop on Uncertainty and Robustness in Deep Learning, 2021.

Zhang, Y., Freedman, Z. B., Hartemink, A. E., Whitman, T., and Huang, J.: Characterizing soil microbial properties using MIR spectra across 12 ecoclimatic zones (NEON sites), Geoderma, 409, 115647, https://doi.org/10.1016/j.geoderma.2021.115647, 2022.

**Comment 20; L. 255:**

MPIW instead of PIW.

**Reply:** Thank you. We corrected it accordingly.

**Comment 21; L. 344:**

"The authors benefited from the shared code of Daniel Bethell. The manuscript would become much more impactful for the soil community if the authors shared their code as well. This would increase the usability of the MC-CP method. Given that some KSSL data has been published in OSSL, it would also be easy to reproduce the study."

**Reply:** Thank you for the suggestion. We will make the code publicly available with the publication.

---

## Author Comment (AC2)

**Referee #2**

General Revision Summary

The study presents a well-executed analysis of uncertainty quantification in deep learning soil spectral models, specifically through the use of Monte Carlo-Conformal Prediction (MC-CP). The paper makes a strong contribution to the field by addressing a crucial gap in soil spectroscopy—reliable uncertainty quantification. The methodological approach is well-documented, and the comparison between MC dropout, Conformal Prediction (CP), and MC-CP is insightful and thorough.

**Reply:** Thank you for your positive comments.

Strengths of the Paper

- Novel Contribution: The paper introduces MC-CP as an method for uncertainty quantification in deep learning soil spectral models. The demonstration of MC-CP's ability to balance expected coverage, computational efficiency, and adaptability to out-of-domain samples is a significant advancement.

- Well-Designed Comparison: The comparison between MC dropout, CP, and MC-CP is informative and shows the trade-offs between these methods.

- Strong Methodological Foundation: The study follows a solid methodological framework.

- Practical Relevance: The application of the proposed method to real-world soil spectral data enhances the practical impact of the study

**Reply:** Thank you for your positive comments.

General Improvements

Terminology and Consistency (Machine Learning vs. Deep Learning)

- The abstract and introduction interchangeably refer to Machine Learning (ML) and Deep Learning (DL). However, the methodology and model used are specifically deep learning-based. Ensure consistency in terminology and explicitly state where ML is a broader category and where DL is specifically applied.

**Reply:** Thank you. We added text to state that ML is the broader category, and the specific method used in this study is for deep learning.

"Additionally, deep learning (DL), as a branch of machine learning, is increasingly being applied to soil science to explore its ability to extract information from large datasets."

"This study introduces an innovative application of Monte Carlo conformal prediction (MC-CP) to quantify uncertainty in deep learning models for predicting clay content from mid-infrared spectroscopy."

"An alternative method to evaluate model uncertainty in DL is the Monte Carlo dropout (MC dropout) by Gal and Ghahramani (2016)"

"This study aimed to explore the use of MC-CP as a potential method to quantify the uncertainty of DL models in soil spectroscopy."

- Incorporate a broader range of examples in the Introduction for Monte Carlo (MC) Dropout and Conformal Prediction (CP), as the current section focuses too narrowly on just two detailed examples.

**Reply:** We added an example by Singh et al. (2024) using conformal prediction for earth observation data.

"Singh et al. (2024) applied CP with ML in earth observation data, and CP successfully generated prediction intervals of canopy height."

Reference:

Singh, G., Moncrieff, G., Venter, Z., Cawse-Nicholson, K., Slingsby, J., and Robinson, T. B.:

Uncertainty quantification for probabilistic machine learning in earth observation using conformal prediction, Scientific Reports, 14(1), 16166, https://doi.org/10.1038/s41598-024-65954-w, 2024.

- There is a lack of clear structure (detailed further in the comments). There are many redundant repetitions, and subordinate clauses with very general information are interspersed throughout, often repeating details that were already mentioned earlier.

**Reply:** We address the issues in the detailed comments section.

- Ensure there is a space between the number and the percentage symbol for proper formatting.

**Reply:** We corrected it accordingly.

Detailed Comments

**Comment 1; L7-10**

"While machine learning has made remarkable strides in predicting various physiochemical properties of soils using spectroscopy, predictions devoid of quantified uncertainty offer limited utility in guiding critical decisions. However, uncertainty quantification remains underutilised in the reporting of soil spectral models, with existing methods facing significant limitations."

The sentence effectively explains that predictions without uncertainty are not useful for decision-making and that uncertainty quantification is rarely used due to limitations. However, the logical connection between these points could be clearer to improve readability and coherence.

Reply: Thank you. We revised the sentences to improve readability and coherence.

"While machine learning has made remarkable strides in predicting various physiochemical properties of soils using spectroscopy, its practical utility in decision-making remains limited without quantified uncertainty. Despite its importance, uncertainty quantification is rarely incorporated into soil spectral models, with existing methods facing significant limitations."

**Comment 2; L10**

" These approaches are either computationally demanding…."

It is not entirely clear whether this refers to the existing methods mentioned in the previous sentence or to something else, as methods and approaches are not necessarily the same.

**Reply:** We revised the sentence as follows:

"Existing methods are either computationally demanding, fail to achieve the desired coverage of observed data, or struggle to handle out-of-domain uncertainty."

**Comment 3; L11-23**

The structure is confusing in the sense that your method is mentioned without prior explanation, followed by the introduction of two established methods for comparison. Additionally, while introducing these methods, you already include some results. To improve clarity, consider restructuring the section by clearly separating the description of methods, the comparison, and then presenting the results.

**Reply:** Thank you for the suggestions. We re-organise the section to separate the methods and results.

"We compared MC-CP with two established methods: (1) Monte Carlo dropout and (2) conformal prediction. Monte Carlo dropout generates prediction intervals for each sample and can address larger uncertainties associated with out-of-domain data. Conformal prediction, on the other hand, guarantees ideal coverage of true values but generates unnecessarily wide

prediction intervals, making it overly conservative for many practical applications. Using 39,177 samples from the mid-infrared spectral library of the Kellogg Soil Survey Laboratory to build convolutional neural networks, we found that Monte Carlo dropout itself falls short in achieving the desired coverage – its 90 % prediction intervals only covered the observed values in 74 % of cases, well below the expected 90 % coverage. In contrast, MC-CP successfully combines the strengths of both methods. It achieved a prediction interval coverage probability of 91 %, closely matching the expected 90 % coverage and far surpassing the performance of the Monte Carlo dropout."

**Comment 4; L24-26**

"This breakthrough enhances the real-world applicability of soil spectral models and represents a significant advancement in the field of soil science. [….] further revolutionising decision-making and risk assessment in soil science."

Shorten this section to two sentences, as the usefulness is stated twice. Avoid redundant explanations to improve clarity.

**Reply:** We revised the sentences as follows:

"The success of MC-CP enhances the real-world applicability of soil spectral models, paving the way for their integration into large-scale machine-learning models, such as soil inference systems, and further transforming decision-making and risk assessment in soil science."

**Comment 5; L29**

"[...](Padarian et al., 2020; Minasny et al., 2024). These studies are characterised by the use of large soil datasets and require an efficient way of extracting information to predict target attributes."

The reference is incorrect, as these studies do not discuss what you describe in the following

sentence.

**Reply:** We changed the references to Ng et al. (2019) and Wadoux et al. (2020) as they discussed the applications of machine learning in (1) soil spectroscopy using a large spectral dataset and (2) digital soil mapping with various data sizes.

Reference:

Ng, W., Minasny, B., Montazerolghaem, M., Padarian, J., Ferguson, R., Bailey, S., and McBratney, A. B.: Convolutional neural network for simultaneous prediction of several soil properties using visible/near-infrared, mid-infrared, and their combined spectra, Geoderma, 352, 251-267, https://doi.org/10.1016/j.geoderma.2019.06.016, 2019.

Wadoux, A. M. J. C., Minasny, B., and McBratney, A. B.: Machine learning for digital soil mapping: Applications, challenges and suggested solutions, Earth-Sci. Rev., 210, 103359, https://doi.org/10.1016/j.earscirev.2020.103359, 2020.

**Comment 6; L41-46**

There are repetitions in the sentences without adding new content. Shorten them for conciseness.

**Reply:** We shorten the sentences as follows:

"Despite the significant success of machine learning in predicting soil properties, uncertainty quantification of the prediction remained an underexplored area in soil spectroscopy (Omondiagbe et al., 2024). The growing demand for practical applications of soil spectral models requires users to know the uncertainty accompanying the model prediction to assess the quality of the predictions (Bellon-Maurel et al., 2010)."

**Comment 7; L43**

"Despite the significant success of machine learning in predicting soil properties, uncertainty

quantification of the prediction remained an underexplored area in soil spectroscopy, and only

a few studies have tried to include uncertainty in the model evaluation."

A reference is needed for the studies mentioned.

**Reply:** We added reference Omondiagbe et al. (2024).

Reference:

Omondiagbe, O. P., Roudier, P., Lilburne, L., Ma, Y., and McNeill, S.: Quantifying uncertainty

in the prediction of soil properties using mid-infrared spectra, Geoderma, 448, 116954,

https://doi.org/10.1016/j.geoderma.2024.116954, 2024.

**Comment 8; L50-54**

I don't see the relevance of explaining the difference between the two types of uncertainty here,

as it does not appear to be a topic in the methods section or the discussion.

**Reply:** Thank you for the suggestions. We remove this part about types of uncertainty.

**Comment 9; L61-66**

To my knowledge, bootstrapping is typically used for confidence intervals, not for prediction

intervals like MC and CP. Additionally, different methods of quantile regression and Gaussian

methods are missing, which would help provide a more complete introduction.

**Reply:** We agree. We revise our content to address these issues.

1.  We added contents to indicate that bootstrapping is only used for confidence intervals

    instead of prediction intervals.

    "In addition, bootstrapping primarily addresses the model uncertainty and it derives

    confidence intervals rather than prediction intervals (Heuvelink, 2014; Wadoux, 2019)."

2.  We added introduction about quantile regression and Bayesian CNNs to provide a more

    complete introduction.

"The diverse nature of models enabled the development of different methods. For example, quantile regression (QR) uses a set of regression models to estimate the quantile of target variables, and the prediction interval can later be defined by the upper and lower quantiles (Kasraei et al., 2021). Additionally, quantile regression forests (QRF) and quantile regression neural networks (QRNN) are also extensions of quantile regression that apply similar principles to generate prediction intervals (Schmidinger and Heuvelink, 2023). Heuvelink et al. (2021) utilised QRF to predict the SOC for soils in Argentina with quantified uncertainty, and the 0.05 and 0.95 quantiles were used to generate the 90 % prediction interval. On the other hand, Omondiagbe et al. (2024) compared bootstrapped PLS, generalised additive models (GAM), and Bayesian CNNs for their ability to quantify uncertainty. They found that GAM and Bayesian CNN outperformed bootstrapped PLS by having PICP close to the ideal 90% value. Moreover, the MPIW of Bayesian CNN is mostly lower than that of GAM models, suggesting a more accurate estimation of uncertainty (Omondiagbe et al., 2024). However, Bayesian neural networks are more intensive in computation compared to standard CNNs (Bethell et al., 2024; Omondiagbe et al., 2024)."

References:

Heuvelink, G. B.: Uncertainty quantification of GlobalSoilMap products, GlobalSoilMap: Basis of the global spatial soil information system, 335-340, 2014.

Heuvelink, G. B. M., Angelini, M. E., Poggio, L., Bai, Z., Batjes, N. H., van den Bosch, R., Bossio, D., Estella, S., Lehmann, J., Olmedo, G. F., and Sanderman, J.: Machine learning in space and time for modelling soil organic carbon change, Eur. J. Soil Sci., 72(4), 1607-1623, https://doi.org/10.1111/ejss.12998, 2021.

Kasraei, B., Heung, B., Saurette, D. D., Schmidt, M. G., Bulmer, C. E., and Bethel, W.: Quantile regression as a generic approach for estimating uncertainty of digital soil maps produced from machine-learning, Environmental Modelling & Software, 144, 105139,

https://doi.org/10.1016/j.envsoft.2021.105139, 2021.

Omondiagbe, O. P., Roudier, P., Lilburne, L., Ma, Y., and McNeill, S.: Quantifying uncertainty in the prediction of soil properties using mid-infrared spectra, Geoderma, 448, 116954, https://doi.org/10.1016/j.geoderma.2024.116954, 2024.

Wadoux, A. M. J. C.: Using deep learning for multivariate mapping of soil with quantified uncertainty, Geoderma, 351, 59-70, https://doi.org/10.1016/j.geoderma.2019.05.012, 2019.

**Comment 10; L68-72**

Specify that MC is specifically used for deep learning to avoid ambiguity.

**Reply:** We added "in DL" to specify that MC is used for deep learning.

"An alternative method to evaluate model uncertainty in DL is the Monte Carlo dropout (MC dropout) by Gal and Ghahramani (2016)"

**Comment 11; L96-103**

"In this study, we applied a strategy to increase the PICP of MC dropout while maintaining its advantages in characterising out-of-domain uncertainty. Monte Carlo-Conformal Prediction (MC-CP) was introduced by Bethell et al. (2024). MC-CP integrates the strengths of both MC dropout and CP."

Clarify that MC-CP is the strategy. Again, avoid repetition to improve clarity and conciseness.

**Reply:** We revised the sentences to improve clarity and conciseness.

"In this study, we applied the Monte Carlo-Conformal Prediction (MC-CP), a method introduced by Bethell et al. (2024) to improve the PICP of MC dropout while maintaining its advantages in characterising out-of-domain uncertainty."

**Comment 12; L113-115**

Please specify how many of the removed samples were due to SOC and how many were excluded because of extreme values.

**Reply:** We revised the sentences to clarify how many samples were excluded in each step.

"The database contains 45,339 samples which have measured MIR spectra and particle size analysis. Since the spectra of mineral and organic soils behave differently, samples with SOC > 10 % were excluded, resulting in the removal 1,808 samples. Additionally, extreme values for clay content were also filtered by excluding data below the 5th percentile and above the 95th percentile, further removing 4,354 samples. This resulted in a total number of 39,177 samples."

**Comment 13; L116**

Clarify why the threshold of 40% clay content was chosen and provide justification for this choice.

**Reply:** We intended to create a subset of soil samples that are spectrally different from the training data. There are two reasons for choosing the 40% threshold. (1) A clay content of 40% is the minimum required for a soil to be classified as clay in most of the soil texture classification system. (2) This threshold results in an in-domain to out-of-domain sample size ratio of 10:1, which is ideal for the analysis. We updated the context to elaborate this:

"A clay content of 40 % is the minimum threshold for a soil to be classified as "clay" according to the US Department of Agriculture soil texture classification (Soil Science Division Staff, 2017). Using this criterion, approximately 10% of the samples were categorised as out-of-domain (clay>40 %, N=3,686), while the remaining in-domain samples had clay content below 40% (N=35,491)."

**Comment 14; L119**

If you are already describing your training and test scheme here, also include the ratio of the splitting mentioned in L203 for consistency and completeness.

**Reply:** Thank you for the suggestion. We moved L203 here to indicate the data split during modelling.

"The in-domain data were further randomly separated into 85 % training, 5 % validation, 5 % calibration for conformal prediction, and 5 % testing."

**Comment 15; Chapter 2.2, 2.3, 2.4**

For better structure, I suggest organizing the section as follows: 2.2 Methods, with subsections 2.2.1 Monte Carlo Dropout (MC dropout), 2.2.2 Conformal Prediction (CP), and 2.2.3 Monte Carlo-Conformal Prediction (MC-CP).

**Reply:** We agree and organised it accordingly as 2.2 Uncertainty quantification methods, 2.2.1 Monte Carlo dropout (MC dropout), 2.2.2 Conformal prediction (CP), and 2.2.3 Monte Carlo-conformal prediction (MC-CP).

**Comment 16; L125**

Missing abbreviation: MC dropout

**Reply:** We corrected it accordingly.

**Comment 17; L128**

"In each dropout layer, a certain portion of the neurons is randomly deactivated (weights set to zero) during both training and testing."

As far as I know, and as stated in the paper by Gal and Ghahramani (2016), neurons are only deactivated during training. While validation can be involved, a specific reason is needed for doing so. Please verify what is happening in your specific use case.

**Reply:** In the MC dropout case, the neurons are deactivated both during training and predicting. Here we refer to the paper from Bethell et al. (2024): *"Although dropout is typically used during training, MC dropout keeps this feature active during inference and performs several forward passes to devise a prediction distribution."* That is, the neurons are deactivated during training and testing.

**Comment 18; L137**

Check the Mathematical notation and terminology of the journal:

https://publications.copernicus.org/for_authors/manuscript_preparation.html#math.

I recommend centering the equations for better readability. Additionally, equations should be treated as nouns within the text. So here I would change it to the following:

The 90% prediction interval […] of the predictions (Eq. 1):

$$Formula. \qquad (Eq. 1)$$

**Reply:** We revised it accordingly.

**Comment 19; L137**

When using a formula, ensure that every abbreviation is defined either before or in the sentence following it. In this case, CMC and Xi are missing definitions.

**Reply:** We revised it accordingly.

"In Eq. 1, $X_i$ represents an individual input sample. The 90 % prediction interval of MC dropout ($C_{MC, 90}$) of each sample $i$ would be defined by the $5^{th}$ quantile ($\hat{q}_5(X_i)$) and the $95^{th}$ quantile ($\hat{q}_{95}(X_i)$) of the predictions (Eq. 1)"

**Comment 20; L150**

Table 1

Stay consistent in using X or Xi throughout the table to maintain clarity and uniformity.

**Reply:** We revised it accordingly.

| $N$ | $f(X_i)$ | $Y_i$ | $r = \|f(X_i) - Y_i\|$ (Nonconformity scores) | $C(X_i) = [f(X_i) - \hat{q}, f(X_i) + \hat{q}]$ |
|---|---|---|---|---|
| 1 | 96 | 95.9 | 0.1 | [93.8, 98.2] |
| 2 | 3 | 3.2 | 0.2 | [0.8, 5.2] |
| 3 | 96 | 95.7 | 0.3 | [93.8, 98.2] |
| 4 | 18 | 18.4 | 0.4 | [15.8, 20.2] |
| 5 | 71 | 70.5 | 0.5 | [68.8, 73.2] |
| 6 | 99 | 99.6 | 0.6 | [96.8, 101.2] |
| 7 | 38 | 37.3 | 0.7 | [35.8, 40.2] |
| 8 | 11 | 11.8 | 0.8 | [8.8, 13.2] |
| 9 | 74 | 73.1 | 0.9 | [71.8, 76.2] |
| 10 | 54 | 55 | 1.0 | [51.8, 56.2] |
| | | | … | |
| 91 | 24 | 21.9 | 2.1 | [21.8, 26.2] |
| 92 | 56 | 58.2 | **2.2** | [53.8, 58.2] |
| 93 | 48 | 45.5 | 2.5 | [45.8, 50.2] |
| 94 | 19 | 21.8 | 2.8 | [16.8, 21.2] |
| 95 | 90 | 86.9 | 3.1 | [87.8, 92.2] |
| 96 | 27 | 30.2 | 3.2 | [24.8, 29.2] |
| 97 | 70 | 66.6 | 3.4 | [67.8, 72.2] |
| 98 | 66 | 69.9 | 3.9 | [63.8, 68.2] |
| 99 | 21 | 16.8 | 4.2 | [18.8, 23.2] |
| 100 | 80 | 84.5 | 4.5 | [77.8, 82.2] |

**Comment 21; L161**

See comment No. 18

**Reply:** We revised it accordingly.

**Comment 22; L170**

Stay consistent in the writing of Monte Carlo-conformal prediction. Since it is based on Bethell et al. (2024), I recommend following their terminology and formatting.

**Reply:** Thank you, we checked the terminology in the article. We chose to use "Monte Carlo-Conformal Prediction (MC-CP)" as the name for the method throughout the manuscript, as this name was also used by Bethell et al. (2024) in their publication. The reason for choosing

"Monte Carlo-Conformal Prediction" instead of "Conformalised Monte Carlo Prediction" is because the former can easily be understood as the combination of MC and CP.

**Comment 23; L179**

See comment No. 18

**Reply:** We revised it accordingly.

**Comment 24; L184**

See comment No. 18

**Reply:** We revised it accordingly.

**Comment 25; L208-209**

See comment No. 18 and a reference is missing for the Eq. 5 and 6.

**Reply:** We revised it accordingly and added reference Ng et al. (2022).

Reference:

Ng, W., Minasny, B., Jeon, S. H., and McBratney, A.: Mid-infrared spectroscopy for accurate measurement of an extensive set of soil properties for assessing soil functions, Soil Secur., 6, 100043, https://doi.org/10.1016/j.soisec.2022.100043, 2022.

**Comment 26; L210-214**

See comment No. 18 a space is missing in Eq. 7 between the fraction and "count".

**Reply:** We revised it accordingly and added a space. Eq. 7 is now $PICP = \frac{1}{n} \, count \, j$.

**Comment 27; L223**

I would rephrase it as follows, omitting the word "poor":

"A negative R-squared value indicates that the model performs worse than simply using the mean prediction."

**Reply:** We agree and modified the sentence accordingly.

"For out-of-domain samples, a negative R-squared value indicates that the model performs worse than simply using the mean prediction"

**Comment 28; L224-225**

Connect the two sentences for example as following:

"Such results for out-of-domain samples were expected, as the model did not have any knowledge of soils with clay content larger than 40%, leading most out-of-domain predictions to fall under 40% clay."

**Reply:** Thank you. We revised the sentence.

"Such a result for out-of-domain samples was expected, as the model lacked knowledge of soils with clay content exceeding 40 %, resulting in the most out-of-domain predictions falling below 40 % clay."

**Comment 29; L238**

"When the evaluation of uncertainty is optimal, the expected coverage of a $p\%$

prediction interval is $p\%$ (dotted line in Fig. 3)"

What do you mean by "evaluation of uncertainty"? Please clarify or provide a more precise definition.

**Reply:** We removed the confusing part and retain the part with expected coverage.

"The expected coverage of a p % prediction interval is p %, which is indicated by the dotted line in Fig. 3"

**Comment 30; L255**

MPIW instead of PIW

**Reply:** We corrected accordingly.

**Comment 31; L263**

Table 4

The PICP value for out-of-domain samples is missing and should be included for completeness.

**Reply:** We updated Fig. 3 and Table 4 to present the PICP of out-of-domain samples:

[Figure]

Figure 3: Prediction interval coverage probability (PICP) of in-domain and out-of-domain samples at different prediction intervals for Monte Carlo dropout, conformal prediction, and Monte Carlo-conformal prediction.

Table 4: Results of uncertainty quantification by Monte Carlo dropout, conformal prediction, and Monte Carlo-conformal prediction. PICP stands for prediction interval coverage probability, and MPIW stands for mean prediction interval width.

| Method | 90 % PICP in-domain | MPIW (%) in-domain | 90 % PICP out-of-domain | MPIW (%) out-of-domain |
|---|---|---|---|---|

| | | | | |
|---|---|---|---|---|
| Monte Carlo dropout | 74 % | 5.56 | 11 % | 6.90 |
| Conformal prediction | 91 % | 11.11 | 32 % | 11.11 |
| Monte Carlo-conformal prediction | 91 % | 9.05 | 26 % | 10.43 |

**Comment 32; L276-281**

I do not agree with the strong wording that MC-CP effectively addresses the out-of-domain issue, as the difference in MPIW between in-domain and out-of-domain samples is not significant.

**Reply:** Thank you for the comment. In our study, the MPIW of out-of-domain samples (10.43 %) was different from the MPIW of in-domain samples (9.05 %) by 1.38 %, this was about 15 % (1.38/9.05) increasement of width. We acknowledge that this difference is not as effective as the results in the carbon prediction in Padarian et al. (2022), which has a significant difference between the in-domain MPIW (2.73 %) and out-of-domain MPIW (15.12 %). We agree to avoid using strong words such as "effectively", and we added sentences to acknowledge that the MPIW is still not wide enough to fully address the out-of-domain samples: "However, the extended MPIW was insufficient to fully address the differences between out-of-domain samples and the in-domain training samples, and the 90 % PICP for MC dropout was only 11 %"

Additionally, we added discussion about the possible reasons behind such performance. In Liu et al. (2021), the authors found that Bayesian NN and MC dropout did not give high uncertainty to out-of-domain samples. Zadorozhny et al. (2021) discussed the problem that sometimes the NN over-generalise from training data to predict out-of-domain samples. That is, when the out-of-domain samples are similar to the in-domain samples, the algorithm wrongly assign high confidence to the prediction of out-of-domain samples. This could be the case in our study, as the spectra of clayey soils were not as distinct from those of sandy soils as the spectra of high

SOC soils were from low SOC soils. The spectra shown in Ng et al. (2022) and Zhang et al. (2022) can support this argument as the high SOC data have significant peaks at 2930-2850 cm$^{-1}$ due to alkyl groups in SOM. Additionally, this plot only include soil samples with organic carbon less than 12%, and the difference of MIR spectra between mineral soils and organic soils could be bigger.

[Figure]

Figure from Ng et al. (2022)

[Figure]

Figure from Zhang et al. (2022)

We added discussion about the possible reasons that MC-CP did not generate a wider MPIW: "However, when facing out-of-domain samples, MC-CP achieved only 26 % coverage at the 90 % prediction interval. The MPIW for out-of-domain samples (10.43 %) was 1.38 higher

than that for in-domain samples (9.05 %), representing a 15 % increase in the width. The difference was insufficient to fully account for out-of-domain uncertainty, leading to the low coverage. Similarly, Liu et al. (2021) found that Bayesian neural networks and MC dropout were unable to assign high uncertainty to out-of-domain samples, indicating overconfidence in predicting unknown data. Zadorozhny et al. (2021) also highlighted the tendency of neural networks to overgeneralise from training data when predicting out-of-domain samples, potentially leading to overconfidence. When out-of-domain sample inputs closely resemble in-domain sample inputs, MC dropout may assign similar confidence levels to out-of-domain samples, failing to capture the true uncertainty. The MIR spectra of clayey soils were not as distinct from those of sandy soils as the spectra of high SOC soils were from low SOC soils (Ng et al., 2022; Zhang et al., 2022). For example, peaks at 2930-2850 cm-1 serve as a distinction between mineral soils and organic soils (Tinti et al., 2015; Ng et al., 2022). Thus, the difference between the MPIW of in-domain and out-of-domain samples was not as significant as in the study of Padarian et al. (2022), in which 20% SOC was used as the separation between in-domain and out-of-domain samples."

References:

Liu, Y., Pagliardini, M., Chavdarova, T., and Stich, S. U.: The Peril of Popular Deep Learning Uncertainty Estimation Methods, Proceedings of the Bayesian Deep Learning workshop, NeurIPS 2021, https://doi.org/10.48550/arXiv.2112.05000, 2021.

Ng, W., Minasny, B., Jeon, S. H., and McBratney, A.: Mid-infrared spectroscopy for accurate measurement of an extensive set of soil properties for assessing soil functions, Soil Secur., 6, 100043, https://doi.org/10.1016/j.soisec.2022.100043, 2022.

Zadorozhny, K., Ulmer, D., and Cinà, G.: Failures of Uncertainty Estimation on Out-Of-Distribution Samples: Experimental Results from Medical Applications Lead to Theoretical

Insights, Proceedings of the ICML 2021 Workshop on Uncertainty and Robustness in Deep Learning, 2021.

Zhang, Y., Freedman, Z. B., Hartemink, A. E., Whitman, T., and Huang, J.: Characterizing soil microbial properties using MIR spectra across 12 ecoclimatic zones (NEON sites), Geoderma, 409, 115647, https://doi.org/10.1016/j.geoderma.2021.115647, 2022

**Comment 33; L299-304**

This part should be discussed directly in the uncertainty section rather than in the limitations and future applications section for better coherence.

**Reply:** We agree. We split the discussion about the out-of-domain samples into two parts and have an additional part in 3.2 Uncertainty quantification for better coherence. The added parts can be seen in comment 32.

**Comment 34; L312**

Specify the exact deep learning model used.

**Reply:** We specify that we used convolutional neural networks.

"The study aimed to assess the uncertainty in predicting clay content using convolutional neural networks through three uncertainty quantification techniques"

**Comment 35; L329**

The wording should be revised—for an optimal trade-off, the results need to be more significant.

**Reply:** We changed the sentence to "Provides a balance between MC Dropout and CP."

---

## Author Response (AR2)

Response to the referee comments on the manuscript:

"Using Monte Carlo conformal prediction to evaluate the uncertainty of deep learning soil spectral models"

Written by

Yin-Chung Huang, José Padarian, Budiman Minasny, and Alex B. McBratney

Submitted to SOIL

Manuscript number: egusphere-2024-3703

**Referee #1:**

In my opinion the authors revised the paper accordingly so that it can be published. It is a great contribution for uncertainty quantification with NNs. While I consider it publishable, I still have some minor comments that could be addressed.

**Reply:** We thank the contribution from referees through this reviewing process.

**L. 22 - 23**

" while still addressing the higher uncertainty in out-of-domain samples."

This part of the sentence should be deleted, because it does not do so very effectively (as can be seen from the out of sample PICP).

**Reply:** Done.

**L. 62 - 72**

While I requested that such a section should be added, I feel it is now a bit too long and not well connected with the previous section and following section. It is important to mention QR due to its popularity and that it can be combined with CP. Then it can be discussed that QR is not (yet) available for all different types of NNs, which is why this new framework is

introduced based on MC and that MC, like QR, can be conformalized.

**Reply:** Thank you for the comment. We decided to keep this section as this part was also requested by the other referee in the first review round. This section is meant to provide introduction on other methods to quantify uncertainty, such as QR and Bayesian CNN.

We added a description to mention that QR is not yet available for all DL models and that CP can be integrated with QR and MC dropout.

"However, QR is not yet available for every DL model." In Line 66.

"CP can therefore be integrated with methods such as QR and MC dropout." In Line 94.

**L. 99 - 103.**

While I requested further context about other methods, in my opinion this new section can be deleted. It is not disruptive but also does not add much since this method is not really used. However, the authors also refer to UNEEC in L. 333, so if the authors deem it necessary, it can be kept.

**Reply:** Thank you for the suggestions, we decided to keep it as we also refer to it in discussion.

**L. 106 - 107**

It could be still useful to mention that it is also known as conformalized monte carlo prediction, as it is then easier to find this paper with search algorithms, as this is the more common name.

**Reply:** We add "Also known as Conformalised Monte Carlo Prediction" in Line 108.

**L. 153**

X_i needs to be italic like the other variables.

**Reply:** Done.

**L. 380:**

"address out-of-domain uncertainties." it only does so to a limited degree. Better drop this from the conclusions.

**Reply:** Done.

**L. 390 Data availability**

When the authors publish the code, they should add the GitHub (or other code repository) link in this section and rename it to "Data and code availability"

**Reply:** Done. We added "The code used to perform MC-CP in this study is available via a GitHub repository (https://doi.org/10.5281/zenodo.15401499, Huang, 2025, and https://github.com/LloydYCHuang/Soil-MC-CP)." In Line 393.